# IMPROVED OFF-POLICY REINFORCEMENT LEARNING IN BIOLOGICAL SEQUENCE DESIGN

## ABSTRACT

Designing biological sequences with desired properties is a significant challenge due to the combinatorially vast search space and the high cost of evaluating each candidate sequence. To address these challenges, reinforcement learning (RL) methods, such as GFlowNets, utilize proxy models for rapid reward evaluation and annotated data for policy training. Although these approaches have shown promise in generating diverse and novel sequences, the limited training data relative to the vast search space often leads to the misspecification of proxy for out-of-distribution inputs. We introduce $\delta$-Conservative Search, a novel off-policy search method for training GFlowNets designed to improve robustness against proxy misspecification. The key idea is to incorporate conservativeness, controlled by parameter $\delta$, to constrain the search to reliable regions. Specifically, we inject noise into high-score offline sequences by randomly masking tokens with a Bernoulli distribution of parameter $\delta$ and then denoise masked tokens using the GFlowNet policy. Additionally, $\delta$ is adaptively adjusted based on the uncertainty of the proxy model for each data point. This enables the reflection of proxy uncertainty to determine the level of conservativeness. Experimental results demonstrate that our method consistently outperforms existing machine learning methods in discovering high-score sequences across diverse tasks—including DNA, RNA, protein, and peptide design—especially in large-scale scenarios. The code is available at https://anonymous.4open.science/r/delta_cs-0477.

## 1 INTRODUCTION

Designing biological sequences with desired properties is crucial in therapeutics and biotechnology (Zimmer, 2002; Lorenz et al., 2011; Barrera et al., 2016; Sample et al., 2019; Ogden et al., 2019). However, this task is challenging due to the combinatorially large search space and the expensive and black-box nature of objective functions. Recent advances in deep learning methods for biological sequence design have shown significant promise at overcoming these challenges (Brookes & Listgarten, 2018; Brookes et al., 2019; Angermueller et al., 2020; Jain et al., 2022).

Among various approaches, reinforcement learning (RL), which leverages a proxy model as a reward function, has emerged as one of the successful paradigms for automatic biological sequence design (Angermueller et al., 2020). RL methods have the benefit of exploring diverse sequence spaces by generating sequences token-by-token from scratch, enabling the discovery of novel sequences. They employ deep neural networks as inexpensive proxy models to approximate costly oracle objective functions. The proxy model serves as a reward function for training deep RL algorithms, enabling the policy network to generate high-reward biological sequences. The model can be trained in an active learning manner (Gal et al., 2017), iteratively annotating new data by querying the oracle with points generated using the policy; these iterations are called query rounds. There are two approaches for training the policy in this context: on-policy and off-policy.

DyNA PPO (Angermueller et al., 2020), a representative on-policy RL method for biological sequence design, employs Proximal Policy Optimization (PPO; Schulman et al., 2017) within a proxy-based active training loop. While DyNA PPO has demonstrated effectiveness in various biological sequence design tasks, its major limitation is the limited search flexibility inherent to on-policy methods. It cannot effectively leverage offline data points, like data collected from previous rounds.

Figure 1: The active learning process for biological sequence design with $\delta$-Conservative Search ($\delta$-CS). Starting with high reward sequences from the offline dataset, we inject token-level noise with probability $\delta$, which determines the conservativeness of the search. Then, the GFlowNet policy denoises the masked sequences. Lastly, the GFlowNet policy is trained with new sequences. After policy training, we query a new batch of sequences and update the dataset for the next round.

Conversely, Generative Flow Networks (GFlowNets; Bengio et al., 2021), off-policy RL methods akin to maximum entropy policies (Tiapkin et al., 2024; Deleu et al., 2024), offer diversity-seeking capabilities and flexible exploration strategies. Jain et al. (2022) applied GFlowNets to biological sequence design with additional Bayesian active learning schemes. They leveraged the off-policy nature of GFlowNets by mixing offline datasets with on-policy data during training. This approach provided more stable training compared to DyNA PPO and resulted in better performance.

However, recent studies have consistently reported that GFlowNets perform poorly on long-sequence tasks such as green fluorescent protein (GFP) design (Kim et al., 2023; Surana et al., 2024). We hypothesize that this poor performance stems from the insufficient quality of the proxy model in the early rounds. For example, in the typical benchmark, the proxy is trained with 5,000 sequences, while GFPs have a combinatorial search space of $20^{238}$, which is bigger than $10^{309}$. GFlowNets are capable of generating sequences from scratch and producing novel sequences beyond the data points. However, when the quality of the proxy model is unreliable, novel sequences that are out-of-distribution to the proxy model yield unreliable results (Trabucco et al., 2021; Yu et al., 2021). This motivates us to introduce a conservative search strategy to restrict search space to the neighborhood of the observed data point during RL training and generating sequences to query for the next round.

**Contribution.** In this paper, we propose a novel off-policy search method called $\delta$-Conservative Search ($\delta$-CS), which enables a trade-off between sequence novelty and robustness to proxy mis-specification by using a conservativeness parameter $\delta$. Specifically, we iteratively train a GFlowNet using $\delta$-CS as follows: (1) we inject noise by independently masking tokens in high-score offline sequences with Bernoulli distribution with parameter $\delta$; (2) the GFlowNet policy sequentially denoises the masked tokens; (3) we use these denoised sequences to train the policy. When $\delta = 1$ (full on-policy), this becomes equivalent to generating full sequences from scratch, and when $\delta = 0$ (fully conservative), this reduces to only showing offline sequences in off-policy training. We adaptively adjust $\delta(x; \sigma)$ using the proxy model's uncertainty estimates $\sigma(x)$ for each data point $x$. This approach allows the level of conservativeness to be adaptively adjusted based on the prediction uncertainty of each data point. Figure 1 illustrates the overall procedure of the $\delta$-CS algorithm.

Our extensive experiments demonstrate that $\delta$-CS significantly improves GFlowNets, successfully discovering higher-score sequences compared to existing model-based optimization methods on diverse tasks, including DNA, RNA, protein, and peptide design. This result offers a robust and scalable framework for advancing research and applications in biotechnology and synthetic biology.

## 2 PROBLEM FORMULATION

We aim to discover sequences $x \in \mathcal{V}^L$ that exhibit desired properties, where $\mathcal{V}$ denotes the vocabulary, such as amino acids or nucleotides, and $L$ represents the sequence length, which is usually fixed. The desired properties are evaluated by a black-box oracle function $f : \mathcal{V}^L \to \mathbb{R}$, which evaluates the desired property of a given sequence, such as binding affinity or enzymatic activity. Evaluating $f$ is often both expensive and time-consuming since it typically involves wet-lab experiments or high-fidelity simulations.

Advancements in experimental techniques have enabled the parallel synthesis and evaluation of sequences in batches. Therefore, lab-in-the-loop processes are emerging as practical settings that

enable active learning. Following this paradigm, we perform $T$ rounds of batch optimization, where in each round, we have the opportunity to query $B$ batched sequences to the (*assumed*) oracle objective function $f$. Due to the labor-intensive nature of these experiments, $T$ is typically very small. Following Angermueller et al. (2020) and Jain et al. (2022), we assume the availability of an initial offline dataset $\mathcal{D}_0 = \{(x^{(n)}, y^{(n)})\}_{n=1}^{N_0}$, where $y = f(x)$. The initial number of data points $N_0$ is typically many orders of magnitude smaller than the size of the search space, as mentioned in the introduction. The goal is to discover, after $T$ rounds, a set of sequences that are novel, diverse, and have high oracle function values.

## 3 ACTIVE LEARNING FOR BIOLOGICAL SEQUENCE DESIGN

Following Jain et al. (2022), we formulate an active learning process constrained by a budget of $T$ rounds with query size $B$, is executed through an iterative procedure consisting of three stages with a novel component of $\delta$-Conservative Search ($\delta$-CS) which will be detailed described in Section 4:

**Step A (Proxy Training):** We train a proxy model $f_\phi(x)$ using the offline dataset $\mathcal{D}_{t-1}$ at round $t$.

**Step B (Policy Training with $\delta$-CS):** We train a generative policy $p(x; \theta)$ using the proxy model $f_\phi(x)$ and the dataset $\mathcal{D}_{t-1}$ with $\delta$-CS.

**Step C (Offline Dataset Augmentation with $\delta$-CS):** We apply $\delta$-CS to query batched data $\{x_i\}_{i=1}^B$ to the oracle $y_i = f(x_i)$. Then the offline dataset is augmented as: $\mathcal{D}_t \leftarrow \mathcal{D}_{t-1} \cup \{(x_i, y_i)\}_{i=1}^B$.

The overall algorithm is described in Algorithm 1. In the following subsections, we describe the details of **Step A** and **Step B**.

### 3.1 STEP A: PROXY TRAINING

Following Jain et al. (2022), we train the proxy model $f_\phi$ using the dataset $\mathcal{D}_{t-1}$ by minimizing the mean squared error loss:

$$\mathcal{L}(\phi) = \mathbb{E}_{x \sim P_{\mathcal{D}_{t-1}}(x)} \left[ (f(x) - f_\phi(x))^2 \right], \tag{1}$$

where $\mathcal{D}_t$ is the dataset at active round $t$, augmented with oracle queries. In the initial round ($t = 1$), we use the given initial dataset $\mathcal{D}_0$. See Appendix A.1 for detailed implementation.

### 3.2 STEP B: POLICY TRAINING WITH $\delta$-CS

For policy training, we employ GFlowNets, which aim to produce samples from a generative policy where the probability of generating a sequence $x$ is proportional to its reward, i.e.,

$$p(x; \theta) \propto R(x; \phi) = f_\phi(x) + \kappa \sigma(x). \tag{2}$$

Following Jain et al. (2022), the reward $R(x; \phi)$ is defined as $f_\phi(x) + \kappa \sigma(x)$, which combines the proxy value $f_\phi(x)$ and the uncertainty $\sigma(x)$ in the form of the upper confidence bound (UCB; Srinivas et al., 2010) acquisition function. This approach prioritizes regions with higher uncertainty, enabling us to query them in the next active round. Here, $\kappa$ is a mixing hyperparameter.

**Policy parameterization.** The forward policy $P_F$ generates state transitions sequentially through trajectories $\tau = (s_0 \rightarrow \ldots \rightarrow s_L = x)$, where $s_0 = ()$ represents the empty sequence, and each state transition involves adding a sequence token. The full sequence $s_L = x$ is obtained after $L$ steps, where $L$ is the length of the sequences. The forward policy $P_F(\tau; \theta)$ is a compositional policy defined as

$$P_F(\tau; \theta) = \prod_{i=1}^L P_F(s_i | s_{i-1}; \theta). \tag{3}$$

GFlowNets have a backward policy $P_B(\tau|x)$ that models the probability of backtracking from the terminal state $x$. The sequence $x = (e_1, \ldots, e_L)$ can be uniquely converted into a state transition trajectory $\tau$, where each intermediate state represents a subsequence. In the case of sequences, there is only a single way to backtrack, so $P_B(\tau|x) = 1$. This makes these types of GFlowNets equivalent

to soft off-policy RL algorithms. For example, the trajectory balance (TB) objective of GFlowNets (Malkin et al., 2022) becomes equivalent to path consistency learning (PCL) (Nachum et al., 2017), an entropy-maximizing value-based RL method according to Deleu et al. (2024).

**Learning objective and training trajectories.** The policy is trained to minimize TB loss as follows.

$$\mathcal{L}_{\text{TB}}(\tau;\theta) = \left( \log \frac{Z_\theta P_F(\tau;\theta)}{R(x;\phi)} \right)^2 \tag{4}$$

Usually, GFlowNets training is employed to minimize TB loss with training trajectories $\tau$ on full supports, asymptotically guaranteeing optimality for the distribution:

$$p(x;\theta) \propto R(x;\phi).$$

A key challenge in prior works Jain et al. (2022) is that the proxy model $f_\phi(x)$ often produces highly unreliable rewards $R(x;\phi)$ for out-of-distribution inputs. In our approach, we mitigate this by providing off-policy trajectories within more reliable regions by injecting conservativeness into off-policy search. Therefore, **we minimize TB loss with $\delta$-CS**, which offers controllable conservativeness.

## 4  $\delta$-CS: CONTROLLABLE CONSERVATIVENESS IN OFF-POLICY SEARCH

This section introduces $\delta$-Conservative Search ($\delta$-CS), an off-policy search method that enables controllable exploration through a conservative parameter $\delta$. Here, $\delta$ defines the Bernoulli distribution governing the masking of tokens in a sequence. Our algorithm is conducted by the following steps:

- Sample high-score offline sequences $x \sim P_{\mathcal{D}_{t-1}}(x)$ from the **rank-based reweighted prior**.
- Inject noise by masking tokens into $x$ using the **noise injection policy** $P_{\text{noise}}(\tilde{x} \mid x, \delta)$.
- Denoise the masked tokens using the **denoising policy** $P_{\text{denoise}}(\hat{x} \mid \tilde{x}; \theta)$.

These trajectories are used to update the GFlowNet parameters $\theta$ by minimizing the loss function $\mathcal{L}_{\text{TB}}(\tau;\theta)$. For more details on the algorithmic components of $\delta$-CS and its integration with active learning GFlowNets, see Algorithm 1.

**Rank-based reweighted prior.** First, we sample a reference sequence $x$ from the prior distribution $P_{\mathcal{D}_{t-1}}$. To exploit high-scoring sequences, we employ rank-based prioritization (Tripp et al., 2020).

$$w(x;\mathcal{D}_{t-1}, k) \propto \frac{1}{kN + \text{rank}_{f,\mathcal{D}_{t-1}}(x)}.$$

Here, $\text{rank}_{f,\mathcal{D}_{t-1}}(x)$ is a relative rank of the value of $f(x)$ in the dataset $\mathcal{D}_{t-1}$ with a weight-shifting factor $k$; we fix $k = 0.01$. This assigns greater weight to sequences with higher ranks. Note that this reweighted prior can also be used during proxy training.

**Noise injection policy.** Let $x = (e_1, e_2, \ldots, e_L)$ denote the original sequence of length $L$. We define a noise injection policy where each position $i \in \{1, 2, \ldots, L\}$ is independently masked according to a Bernoulli distribution with parameter $\delta \in [0, 1]$, resulting in the masked sequence $\tilde{x} = (\tilde{e}_1, \tilde{e}_2, \ldots, \tilde{e}_L)$. The noise injection policy $P_{\text{noise}}(\tilde{x} \mid x, \delta)$ is defined as:

$$P_{\text{noise}}(\tilde{x} \mid x, \delta) = \prod_{i=1}^{L} \left[ \delta \cdot \mathbb{I}\{\tilde{e}_i = [\text{MASK}]\} + (1 - \delta) \cdot \mathbb{I}\{\tilde{e}_i = e_i\} \right],$$

where $\mathbb{I}\{\cdot\}$ is the indicator function.

**Denoising policy.** We employ the GFlowNet forward policy $P_F$ to sequentially reconstruct the masked sequence $\tilde{x} = (\tilde{e}_1, \tilde{e}_2, \ldots, \tilde{e}_L)$ by predicting tokens from left to right. The probability of denoising next token $\tilde{e}_t$ from previously denoised subsquence $\hat{s}_{t-1}$ is:

$$P_{\text{denoise}}(\hat{e}_t \mid \hat{s}_{t-1}, \tilde{x}; \theta) = \begin{cases} \mathbb{I}\{\hat{e}_t = \tilde{e}_t\}, & \text{if } \tilde{e}_t \neq [\text{MASK}], \\ P_F(\hat{s}_t = (\hat{s}_{t-1}, \hat{e}_t) \mid \hat{s}_{t-1}; \theta), & \text{if } \tilde{e}_t = [\text{MASK}]. \end{cases}$$

The fully reconstructed sequence $\hat{x} = \hat{s}_L$ is obtained by sampling from:

$$P_{\text{denoise}}(\hat{x} \mid \tilde{x}; \theta) = \prod_{t=1}^{L} P_{\text{denoise}}(\hat{e}_t \mid \hat{s}_{t-1}, \tilde{x}; \theta).$$

By denoising the masked tokens with the GFlowNet policy, which infers each token sequentially from left to right, we generate new sequences $\hat{x}$ that balance novelty and conservativeness through the parameter $\delta$.

### 4.1 Adjusting conservativeness parameter $\delta$

Determining the conservative parameter $\delta$ is a crucial aspect of the algorithm. We propose and study two variants, constant and adaptive $\delta$.

**Constant.** As a simple approach, we set $\delta$ as a constant, selecting it to have the noise policy mask 4–15 tokens per sequence. Despite its simplicity, this choice effectively enhances policy training and leads to the discovery of high-scoring sequences during active rounds; we provide further studies on $\delta$-CS with a constant $\delta$ in Appendix B.1.

**Adaptive.** Another intuitive approach is to adjust $\delta$ based on the uncertainty of the proxy $\sigma$ on each sequence $x$, that is $\delta(x; \sigma)$. Specifically, we define a function that assigns lower $\delta$ values for highly uncertain samples and vice versa: $\delta(x; \sigma) = \delta_{\text{const}} - \lambda\sigma(x)$. We estimate $\sigma(x)$, the standard deviation of the proxy model $f_\phi(x)$, via MC dropout (Gal & Ghahramani, 2016) or an ensemble method (Lakshminarayanan et al., 2017). $\lambda$ is a scaling factor and related to the influence of the proxy uncertainty on $\delta$; we set it to satisfy $\lambda\mathbb{E}_{P_{\mathcal{D}_0}(x)}\sigma(x) \approx \frac{1}{L}$ based on the observations from the initial round.

In our main experiments, we use adaptive $\delta(x, \sigma)$ as the default setup.

## 5 Related work

### 5.1 Biological Sequence Design

Designing biological sequences using machine learning methods is widely studied. Bayesian optimization (BO) methods (Mockus, 2005; Belanger et al., 2019; Zhang et al., 2022) exploit posterior inference over newly acquired data points to update a Bayesian proxy model that can measure useful uncertainty. The BO method can be greatly improved in high-dimensional tasks by using trust-region-based search restrictions (Wan et al., 2021; Eriksson et al., 2019; Biswas et al., 2021; Khan et al., 2023) and by combining it with deep generative models (Stanton et al., 2022; Gruver et al., 2024). However, these methods usually suffer from scalability issues due to the complexity of the Gaussian process (GP) kernel (Belanger et al., 2019) or the difficulty of sampling from an intractable posterior (Zhang et al., 2022).

Offline model-based optimization (MBO) (Kumar & Levine, 2020; Trabucco et al., 2021; Yu et al., 2021; Chen et al., 2022; Kim et al., 2023; Chen et al., 2023a; Yun et al., 2024) also addresses the design of biological sequences using offline datasets only, which can be highly efficient because they do not require oracle queries. These approaches have reported meaningful findings, such as the conservative requirements on proxy models since proxy models tend to give high rewards on unseen samples (Trabucco et al., 2021; Yu et al., 2021; Yuan et al., 2023; Chen et al., 2023b). This supports our approach of adaptive conservatism in the search process. Surana et al. (2024) recently noted that offline design and existing benchmarks are insufficient to reflect biological reliability, indicating that settings without additional Oracle queries might be too idealistic.

Reinforcement learning methods, such as DyNA PPO (Angermueller et al., 2020) and GFlowNets (Bengio et al., 2021; Jain et al., 2022; 2023b; Hernández-García et al., 2024), and sampling with generative models (Brookes & Listgarten, 2018; Brookes et al., 2019; Das et al., 2021; Song & Li, 2023) aim to search the biological sequence space using a sequential decision-making process with a policy, starting from scratch. Similarly, sampling with generative models (Brookes & Listgarten, 2018; Brookes et al., 2019; Song & Li, 2023) searches the sequence space using generative models like VAE (Kingma & Welling, 2014). While these approaches allow for the creation of novel

sequences, as sequences are generated from scratch, they are relatively prone to incomplete proxy models, particularly in regions where the proxy is misclassified due to being out-of-distribution.

An alternative line of research is evolutionary search (Arnold, 1998; Bloom & Arnold, 2009; Schreiber et al., 2020; Sinai et al., 2020; Ren et al., 2022; Ghari et al., 2023; Kirjner et al., 2024), a popular method in biological sequence design. Especially Ghari et al. (2023) proposed GFNSeqEditor, which utilizes GFlowNets as prior distribution to edit biological sequences as an evolutionary search. Evolutionary search methods iteratively edit given sequences and constrain the new sequences so as not to deviate too far from the seed sequence; they usually start from the *wild-type*, which occurs in nature. This can be viewed as constrained optimization, where out-of-distribution for the proxy model can lead to unrealistic and low-score biological sequences. Consequently, they do not aim to produce highly novel sequences. Our method can be seen as a hybrid of off-policy RL and evolutionary search, capitalizing on both the high novelty offered by GFlowNets and the high rewards with out-of-distribution robustness provided by constrained search where they are properly balanced by $\delta$. Our experimental comparison with GFNSeqEditor (Ghari et al., 2023) demonstrates this hybridization balanced by $\delta$ enables us to discover sequences with greater novelty and higher rewards rather than merely using the GFlowNet policy as editing priors.

## 5.2 GFLOWNETS

GFlowNets were introduced by Bengio et al. (2021) and unified by Bengio et al. (2023), demonstrating effectiveness across various domains, including language modeling (Hu et al., 2023), diffusion models (Sendera et al., 2024; Venkatraman et al., 2024), and scientific discovery (Jain et al., 2022; 2023a). Several works have aimed to improve their training methods for better credit assignment (Malkin et al., 2022; Madan et al., 2023; Pan et al., 2023; Jang et al., 2024) and extensions to multi-objective settings (Jain et al., 2023b; Chen & Mauch, 2024). Orthogonal to this, researchers have investigated better off-policy exploration methods (Rector-Brooks et al., 2023; Shen et al., 2023; ?; Kim et al., 2024c;a;b). Our method is particularly related to these exploration methods, yet the major difference is that they are designed under the assumption that the reward model is accurate, which does not hold in active learning and thus requires conservativeness.

## 6 EXPERIMENTS

Following the FLEXS benchmark (Sinai et al., 2020),[1] we evaluate our proposed method on various biological sequence design tasks. Furthermore, we analyze the effect of $\delta$-CS by directly comparing with GFN-AL on TF-Bind-8 and an anti-microbial peptide design in Section 6.6. For each experiment, we conduct five independent runs.

**Implementation details.** For proxy models, we employ a convolutional neural network (CNN) with one-dimensional convolutions (Sinai et al., 2020) with a UCB acquisition function and an ensemble of three network instances to measure the uncertainty. Note that we use the same architecture to implement proxy models for all baselines. For the GFlowNet policy, we use a simple two-layer long short-term memory (LSTM) network (Hochreiter & Schmidhuber, 1997) and train the policy with 1,000 inner-loop updates using a learning rate of $5 \times 10^{-4}$ with a batch size of 256. However, in TF-Bind-8 and AMP, where we analyze the effectiveness of $\delta$-CS compared to GFN-AL, we directly implement $\delta$-CS on top of the GFN-AL implementation. Lastly, we set $\delta$ and $\lambda$ according to the description in Section 4; specifically, $\delta = 0.5$ for tasks with $L \leq 50$ and $\delta = 0.05$ for long-sequences. More details are provided in Appendix A.2.

**Baselines.** As our baseline methods, we employ representative exploration algorithms. Further details are provided in Appendix A.3.

- **AdaLead** (Sinai et al., 2020) is a well-implemented model-guided evaluation method with a hill-climbing algorithm.
- **Bayesian optimization** (BO; Snoek et al., 2012) is a classical algorithm for black-box optimization. We employ the BO algorithm with a sparse sampling of the mutation space implemented by Sinai et al. (2020).

---

[1]FLEXS (Fitness Landscape EXploration Sandbox) is a widely-used open-source simulation environment for biological sequence design, which is available at https://github.com/samsinai/FLEXS.

Table 1: Maximum rewards achieved by each baseline method across six tasks, with the sequence length ($L$) for each task specified. The mean and standard deviation from five runs are reported. The highest values for each task are highlighted in **bold**.

| | RNA-A ($L = 14$) | RNA-B ($L = 14$) | RNA-C ($L = 14$) | TF-Bind-8 ($L = 8$) | GFP ($L = 238$) | AAV ($L = 90$) |
|---|---|---|---|---|---|---|
| AdaLead | $0.968 \pm 0.070$ | $0.965 \pm 0.033$ | $0.867 \pm 0.081$ | $\mathbf{0.995 \pm 0.004}$ | $3.581 \pm 0.004$ | $0.565 \pm 0.027$ |
| BO | $0.722 \pm 0.025$ | $0.720 \pm 0.032$ | $0.694 \pm 0.034$ | $0.977 \pm 0.008$ | $3.572 \pm 0.000$ | $0.500 \pm 0.000$ |
| CMA-ES | $0.816 \pm 0.030$ | $0.850 \pm 0.063$ | $0.753 \pm 0.062$ | $0.986 \pm 0.008$ | $3.572 \pm 0.000$ | $0.500 \pm 0.000$ |
| CbAS | $0.678 \pm 0.020$ | $0.668 \pm 0.021$ | $0.696 \pm 0.041$ | $0.988 \pm 0.004$ | $3.572 \pm 0.000$ | $0.500 \pm 0.000$ |
| DbAS | $0.670 \pm 0.041$ | $0.652 \pm 0.021$ | $0.678 \pm 0.025$ | $0.987 \pm 0.004$ | $3.572 \pm 0.000$ | $0.500 \pm 0.000$ |
| DyNA PPO | $0.737 \pm 0.022$ | $0.730 \pm 0.088$ | $0.728 \pm 0.060$ | $0.977 \pm 0.013$ | $3.572 \pm 0.000$ | $0.500 \pm 0.000$ |
| GFN-AL | $1.030 \pm 0.024$ | $1.001 \pm 0.016$ | $0.951 \pm 0.034$ | $0.976 \pm 0.002$ | $3.578 \pm 0.003$ | $0.560 \pm 0.008$ |
| **GFN-AL + $\delta$-CS** | $\mathbf{1.055 \pm 0.000}$ | $\mathbf{1.014 \pm 0.001}$ | $\mathbf{1.094 \pm 0.045}$ | $0.981 \pm 0.002$ | $\mathbf{3.592 \pm 0.003}$ | $\mathbf{0.708 \pm 0.010}$ |

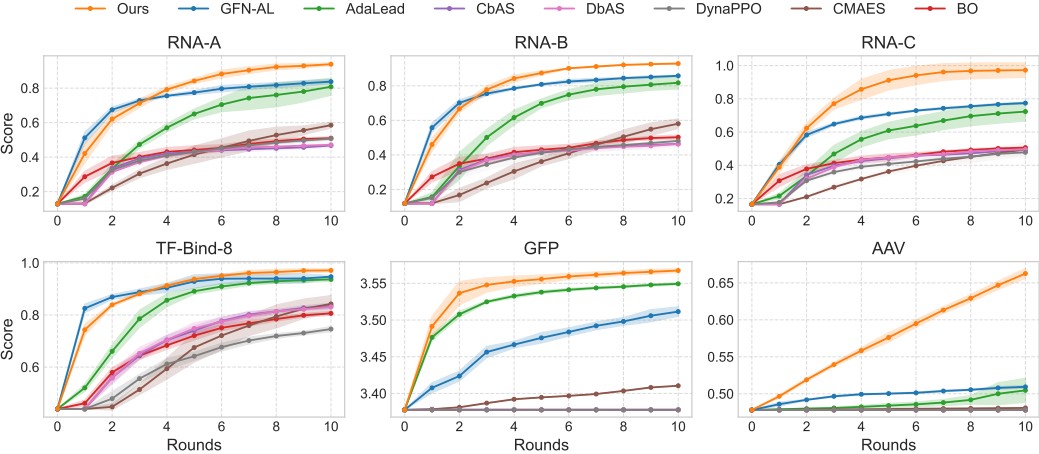

Figure 2: Median scores of Top-128 over active rounds. Ours (GFN-AL + $\delta$-CS) consistently outperforms baseline in RNA, DNA (TF-Bind-8), and protein (GFP and AAV) design tasks.

- **CMA-ES** (Hansen, 2006) is another well-known evolutionary algorithm that optimizes a continuous relaxation of one-hot vectors encoding sequence with the covariance matrix.
- **CbAS** (Brookes et al., 2019) and **DbAS** (Brookes & Listgarten, 2018) are probabilistic frameworks that use model-based adaptive sampling with a variational autoencoder (VAE; Kingma & Welling, 2014). Notably, CbAS restricts the search space with a trust-region search similar to the proposed method.
- **DyNA PPO** (Angermueller et al., 2020) uses proximal policy optimization (PPO; Schulman et al., 2017), an on-policy training method.
- **GFN-AL** (Jain et al., 2022) is our main baseline that uses GFN with Bayesian active learning.

For each task, we conduct 10 active learning rounds starting from the initial dataset $\mathcal{D}_0$. The query batch size is all set as 128 except for the AMP design, whose query size is 1,000. Further details of each task are provided in the following subsections. To evaluate the performance, we measure the maximum, median, and mean scores of Top-$K$ sequences.

## 6.1 RNA SEQUENCE DESIGN

**Task setup.** The goal is to design an RNA sequence that binds to the target with the lowest binding energy, which is measured by ViennaRNA (Lorenz et al., 2011). The length ($L$) of RNA is set to 14, with 4 tokens. In this paper, we have three RNA binding tasks, RNA-A, RNA-B, and RNA-C, whose initial datasets consist of 5,000 randomly generated sequences with certain thresholds; we adopt the offline dataset provided in Kim et al. (2023). We use $\delta = 0.5$ and $\lambda = 5$, according to the guidelines in Section 4.

**Results.** As shown in Figure 2 and Table 1, our method outperforms all baseline approaches. The curve in Figure 2 increases significantly faster than the other methods, indicating that $\delta$-CS effec-

Table 2: Results on AMP with different acquisition functions (UCB, EI). The mean and standard deviation from five runs are reported. Improved results with $\delta$-CS over GFN-AL are marked in **bold**.

| | Max | Mean | Diversity | Novelty |
|---|---|---|---|---|
| COMs | $0.930 \pm 0.001$ | $0.920 \pm 0.000$ | $0.000 \pm 0.000$ | $11.869 \pm 0.298$ |
| DyNA PPO | $0.953 \pm 0.005$ | $0.941 \pm 0.012$ | $15.186 \pm 5.109$ | $16.556 \pm 3.653$ |
| GFN-AL (UCB) | $0.936 \pm 0.004$ | $0.919 \pm 0.005$ | $\mathbf{28.504 \pm 2.691}$ | $19.220 \pm 1.369$ |
| GFN-AL + $\delta$-CS (UCB) | $\mathbf{0.948 \pm 0.015}$ | $\mathbf{0.938 \pm 0.016}$ | $25.379 \pm 3.735$ | $\mathbf{23.551 \pm 1.290}$ |
| GFN-AL (EI) | $0.950 \pm 0.002$ | $0.940 \pm 0.003$ | $15.576 \pm 7.896$ | $21.810 \pm 4.165$ |
| GFN-AL + $\delta$-CS (EI) | $\mathbf{0.962 \pm 0.003}$ | $\mathbf{0.958 \pm 0.004}$ | $\mathbf{16.631 \pm 2.135}$ | $\mathbf{24.946 \pm 4.246}$ |

tively trains the policy and generates appropriate queries in each active round. More results are provided in Appendix C.1.

## 6.2 DNA SEQUENCE DESIGN

**Task setup.** In this task, we aim to generate diverse and novel DNA sequences that maximize the binding affinity to the target transcription factor. The length ($L$) of the sequence is fixed with 8. The initial dataset $\mathcal{D}_0$ is the bottom 50% in terms of the score, which results in $32,898$ samples, with the maximum score of 0.439. Though this has been widely used in many studies (Sinai et al., 2020; Jain et al., 2022; Trabucco et al., 2022; Kim et al., 2023), the TF-Bind-8 is easy to optimize, especially due to its size (Sinai et al., 2020). Similar to RNA, we use $\delta = 0.5$ and $\lambda = 5$.

**Results.** As shown in Table 1, AdaLead achieves the highest maximum performance, while $\delta$-CS still outperforms GFN-AL. We believe that AdaLead's greedy evolutionary search capability is powerful, especially in the small search space of TF-bind-8. However, in Figure 2, $\delta$-CS demonstrates the best median performance compared to the other baselines; see the mean, diversity, and novelty in Appendix C.2.

## 6.3 PROTEIN SEQUENCE DESIGN

We consider two protein sequence design tasks: the green fluorescent protein (GFP; Sarkisyan et al., 2016) and additive adeno-associated virus (AAV; Ogden et al., 2019).

**GFP.** The objective is to identify protein sequences with high log-fluorescence intensity values.[2] The vocabulary is defined as 20 standard amino acids, i.e., $|\mathcal{V}| = 20$, and the sequence length $L$ is 238; thus, we set $\delta$ as 0.05 and $\lambda$ as 0.1, according to our guideline. The initial datasets are generated by randomly mutating the provided wild-type sequence for each task while filtering out sequences that have higher scores than the wild-type; we obtain the initial dataset with $|\mathcal{D}_0| = 10\,200$ with a maximum score value of 3.572.

**AAV.** The aim is to discover sequences that lead to higher gene therapeutic efficiency. The sequences are composed of the 20 standard amino acids with a length of 90, resulting in the search space of $20^{90}$. In the same way as in GFP, we collect an initial dataset of 15,307 sequences with a maximum score of 0.500. We use $\delta = 0.05$ and $\lambda = 1$.

**Results.** Table 1 shows the results of all methods in protein sequence design tasks. Given the combinatorially vast design space with sequence lengths $L = 238$ and 90, most baselines fail to discover new sequences whose score is higher than the maximum of the dataset. In contrast, as depicted in Figure 2, our method finds high-score sequences beyond the dataset, even with a single active round. This underscores the superiority of our search strategy in practical biological sequence design tasks. Full results are provided in Appendix C.3.

## 6.4 ANTI-MICROBIAL PEPTIDE DESIGN

**Task setup.** The goal is to generate protein sequences with anti-microbial properties (AMP). The vocabulary size $|\mathcal{V}| = 20$, and the sequence length ($L$) varies across sequences, and we consider sequences of length 50 or lower. For the AMP task, we consider a much larger query batch size for

---

[2]The score is evaluated by ML oracle models. FLEXS uses TAPE (Rao et al., 2019) for evaluation, while Design Bench Transformer (Trabucco et al., 2022) is employed in GFN-AL.

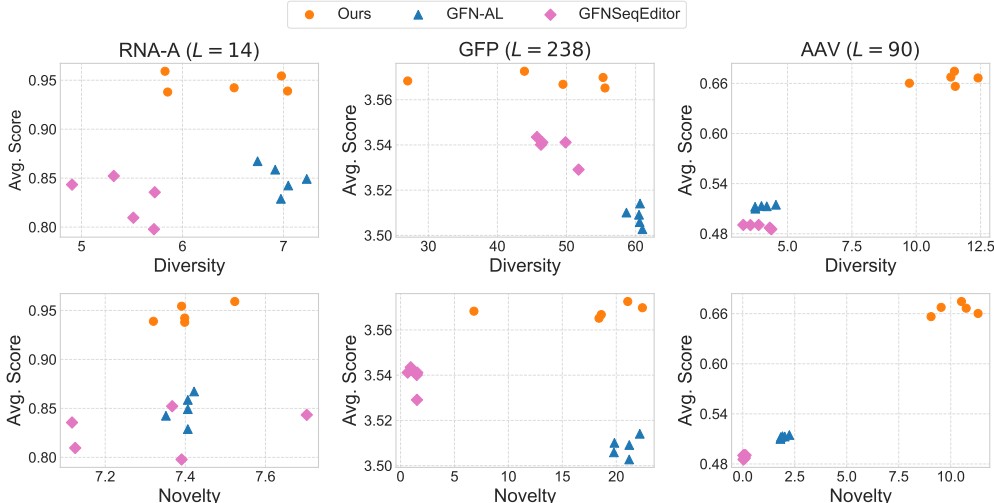

Figure 3: Average score and diversity/novelty with five independent runs. Our method (GFN-AL + $\delta$-CS) consistently approaches Pareto frontier performance. We set $\delta = 0.5$ for short sequences ($L \leq 50$) and set $\delta = 0.05$ for long length sequences ($L > 50$).

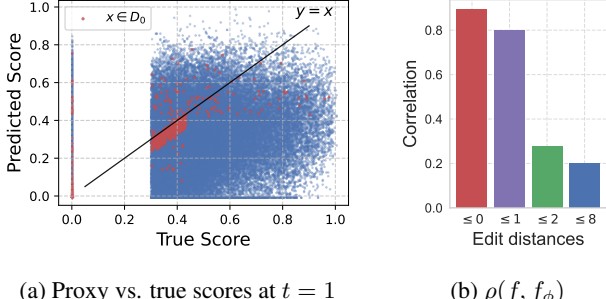

(a) Proxy vs. true scores at $t = 1$         (b) $\rho(f, f_\phi)$

Figure 4: Proxy failure on Hard TF-Bind-8. (a) shows the proxy values (i.e., reward) and true score on the whole data point at the initial round. In (b), the correlation between $f$ and $f_\phi$ is much higher when the data points are close to the initial dataset ('$\leq 0$' and '$\leq 8$' correspond to the initial dataset and the whole sequence space, respectively).

each active round because they can be efficiently synthesized and evaluated (Jain et al., 2022). We set $\delta$ as 0.5 with $\lambda = 1$.

**Results.** The results in Table 2 illustrate that ours consistently gives improved performance over GFN-AL regardless of acquisition function. According to the work from Jain et al. (2022), we also adopt conservative model-based optimization method, (COMs; Trabucco et al., 2021) and on-policy reinforcement learning, DyNA PPO (Angermueller et al., 2020) as baselines. Our method demonstrated significantly higher performance in terms of mean, diversity, and novelty compared to the baselines.

## 6.5 ACHIEVING PARETO FRONTIER WITH BALANCING CAPABILITY OF $\delta$-CS

In this analysis, we demonstrate that $\delta$-CS achieves a balanced search using $\delta$, producing Pareto frontiers or comparable results to the baseline methods: GFN-AL (Jain et al., 2022) and GFNSeqEditor (Ghari et al., 2023). Notably, GFN-AL can be seen as a variant of our method with $\delta = 1$, which fully utilizes the entire trajectory search. This approach is expected to yield high novelty and diversity, but it is also prone to generating low rewards due to the increased risk of out-of-distribution samples affecting the proxy model. GFNSeqEditor, on the other hand, leverages GFlowNets as a prior, editing from a wild-type sequence. It is designed to deliver reliable performance and be more robust to out-of-distribution issues by constraining the search to sequences similar to the wild type. However, unlike $\delta$-CS, GFNSeqEditor does not utilize such obtained samples for training

GFlowNets in full trajectory level; GFNSeqEditor is expected to have lower diversity and novelty compared to GFN-AL and $\delta$-CS.

As shown in Fig. 3, GFN-AL generally produces higher diversity and novelty in the RNA and GFP tasks compared to GFNSeqEditor. However, GFNSeqEditor performs better in terms of reward on the large-scale GFP task, whereas GFN-AL struggles due to the lack of a constrained search procedure in such a large combinatorial space. In contrast, $\delta$-CS achieves Pareto-optimal performance compared to both methods, clearly outperforming GFNSeqEditor across six tasks, with higher rewards, diversity, and novelty. For the RNA and GFP tasks, we achieve higher scores than GFN-AL while maintaining similar novelty but slightly lower diversity. In the AAV task, $\delta$-CS shows a distinct Pareto improvement. These results demonstrate that $\delta$-CS provides a beneficial balance by combining conservative search with amortized inference on full trajectories using off-policy GFlowNets training, effectively capturing the strengths of both GFN-AL and GFNSeqEditor. The results with various $\delta$ are provided in Appendix B.3.

### 6.6 STUDY ON PROXY FAILURE AND CONSERVATIVENESS EFFECT

**Task: Hard TF-Bind-8.** By modifying the initial dataset distribution and the landscape, we can make a harder version of TF-Bind-8. Specifically, we collect the initial dataset near a certain sequence (considered as a wild-type) while ensuring that the initial sequences have lower scores than the given sequence, which is 0.431. The size of $\mathcal{D}_0$ is 1,024. Furthermore, we modify the landscape to give 0 rewards for sequences with scores lower than 0.3. These features are often observed in protein design tasks, where the search space is extremely large, e.g., $20^{238}$ previously–with a limited real-world dataset, and the score often falls to 0.

**Proxy failure.** As shown in Figure 2, $\delta$-CS gives slightly better performance than GFN-AL, but only marginally. This is because the TF-Bind-8 task is relatively easy to optimize, leading to similar results across methods. To more clearly assess the effectiveness of $\delta$-CS, we conduct several studies on the harder TF-Bind-8 task, which is more difficult to optimize. Figure 4a illustrates the proxy values and true scores for all $x \in \mathcal{X}$ in the first round. While the proxy provides accurate predictions for the initial data points $x \in \mathcal{D}_0$ (represented by the red dots), it produces unreliable predictions for points outside $\mathcal{D}_0$. This supports our hypothesis that the proxy model performs poorly on out-of-distribution data.

**Effect of $\delta$ conservativeness.** Figure 4b illustrates that the correlation between the oracle $f$ and the proxy $f_\phi$ significantly decreases as data points move farther from the observed dataset. This strongly motivates the use of $\delta$-CS, which constrains the search bounds using $\delta$. By limiting the search to within these constrained edit distances, $\delta$-CS enhances the correlation with the oracle.

**Studies on $\delta$.** We study on the choice of $\delta$ and effectiveness of adaptive $\delta(x, \sigma)$, in Appendix B.1. Further experimental results, including ablation studies, are provided in Appendix B.

## 7 CONCLUSIONS

In this paper, we introduced a novel off-policy sampling method for GFlowNets, called $\delta$-CS, which provides controllable conservativeness through the use of a $\delta$ parameter. Additionally, we proposed an adaptive conservativeness approach by adjusting $\delta$ for each data point based on prediction uncertainty. We demonstrated the effectiveness of $\delta$-CS in active learning GFlowNets, achieving strong performance across various biological sequence design tasks, including DNA, RNA, protein, and peptide design, consistently outperforming existing baselines.

**Limitations and future works.** The main limitation of our method is that it doesn't fundamentally resolve the drawbacks of active learning; it serves as a useful tool within the existing framework. Investigating robust proxy training and uncertainty measurement techniques remains necessary. These improvements are orthogonal to our approach and can enhance $\delta$-CS when integrated.

Future work includes combining $\delta$-CS with existing GlowNet methods. For example, applying improved credit assignment for larger-scale tasks (Jang et al., 2024) and extending to multi-objective settings (Jain et al., 2023b; Chen & Mauch, 2024) could significantly boost its applicability and effectiveness.

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

# A    IMPLEMENTATION DETAIL

---

**Algorithm 1** Active Learning GFlowNets with $\delta$-CS

---

1: **Input:** Oracle $f$, initial dataset $\mathcal{D}_0$, active rounds $T$, query size $B$, training batch size $2 \times M$.
2: **procedure** $\delta$-CS $(\mathcal{D}_{t-1}, M, \delta)$          $\triangleright$ $\delta$-CS subroutine
3:      sample high reward data $x_1, \ldots, x_M$ with **rank-based reweighed prior** $P_{\mathcal{D}_{t-1}}(\cdot)$.
4:      obtain masked data $\tilde{x}_1, \ldots, \tilde{x}_M$ with **noise injection policy** $P_{\text{noise}}(\tilde{\cdot}|\cdot, \delta)$ from $x_1, \ldots, x_M$.
5:      obtain denoised $\hat{x}_1, \ldots, \hat{x}_M$ with **denoising policy** $P_{\text{denoise}}(\hat{\cdot} \mid \tilde{\cdot}; \theta)$ from $\tilde{x}_1, \ldots, \tilde{x}_M$.
6:      **return** $\hat{x}_1, \ldots, \hat{x}_M$.
7: **end procedure**
8: **for** $t = 1$ to $T$ **do**          $\triangleright$ Active learning with $T$ rounds
9:      **while** proxy training iterations **do**          $\triangleright$ **Step A:** Proxy training
10:         train proxy $f_\phi(x)$ with current round dataset $\mathcal{D}_{t-1}$:

$$\mathcal{L}(\phi) = \mathbb{E}_{x \sim P_{\mathcal{D}_{t-1}}(x)} \left[ (f(x) - f_\phi(x))^2 \right].$$

11:      **end while**
12:      **while** policy training iterations **do**          $\triangleright$ **Step B:** Policy training
13:         obtain off-policy trajectories $\hat{\tau}_1, \ldots, \hat{\tau}_M$ from $\hat{x}_1, \ldots, \hat{x}_M$ given by $\delta$-CS $(\mathcal{D}_{t-1}, M, \delta)$.
14:         obtain offline trajectory $\tau_1, \ldots, \tau_M$ from $x_1, \ldots, x_M \sim P_{\mathcal{D}_{t-1}}(\tau)$.
15:         train $\theta$ with TB loss over $\hat{\tau}_1, \ldots, \hat{\tau}_M$ and $\tau_1, \ldots, \tau_M$

$$\frac{1}{2M} \sum_{i=1}^{M} \left( \log \frac{Z_\theta P_F(\tau_i; \theta)}{R(x_i; \phi)} \right)^2 + \frac{1}{2M} \sum_{i=1}^{M} \left( \log \frac{Z_\theta P_F(\hat{\tau}_i; \theta)}{R(\hat{x}_i; \phi)} \right)^2.$$

16:      **end while**
17:      obtain query samples $\hat{x}_1, \ldots, \hat{x}_B$ from $\delta$-CS $(\mathcal{D}_{t-1}, B, \delta)$.
18:      $\mathcal{D}_t \leftarrow \mathcal{D}_{t-1} \cup \{(\hat{x}_i, f(\hat{x}_i))\}_{i=1}^{B}$.      $\triangleright$ **Step C:** Dataset augmentation with oracle $f$ query
19: **end for**

---

## A.1    PROXY TRAINING

For training proxy models, we follow the procedure of (Jain et al., 2022). We use Adam (Kingma, 2015) optimizer with learning rate $1 \times 10^{-5}$ and batch size of 256. The maximum proxy update is set as 3000. To prevent over-fitting, we use early stopping using the 10% of the dataset as a validation set and terminate the training procedure if validation loss does not improve for five consecutive iterations.

## A.2    POLICY TRAINING

As described in Section 6, we employ a two-layer long short-term memory (LSTM; Hochreiter & Schmidhuber, 1997) with 512 hidden dimensions. The policy is trained with a learning rate of $5 \times 10^{-4}$ with a batch size of 256. The learning rate of $Z$ is set as $10^{-3}$. The coefficient $\kappa$ in Equation (2) is set as 0.1 for TF-Bind-8 and AMP with MC dropout, according to Jain et al. (2022), and 1.0 for RNA and protein design with Ensemble following Ren et al. (2022).

## A.3    IMPLEMENTATION DETAILS OF BASELINES

We adopt open-source code from FLEXS benchmark (Sinai et al., 2020).

- **AdaLead** (Sinai et al., 2020): We use a default settings of hyperparmeters for AdaLead. Specifically, we use a recombination rate of 0.2, mutation rate of $1/L$, where $L$ is sequence length, and threshold $\tau = 0.05$.
- **DbAS** (Brookes & Listgarten, 2018): We implement DbAS with variational autoencoder (VAE; Kingma & Welling, 2014) as the generator. The input is a one-hot encoding vector, and the

output latent dimension is 2. In each cycle, DbAS starts by training the VAE with the top 20% sequences in terms of the score.

- **CbAS** (Brookes et al., 2019): Similar to DbAS, we implement CbAS with VAE. The main difference from DbAS is that we select top 20% sequences with the weights $p(\boldsymbol{x}|\boldsymbol{z}, \theta^{(0)})/q(\boldsymbol{x}|\boldsymbol{z}, \phi^{(t)})$, where $p(\cdot; \theta^{(0)})$ is trained with the ground-truth samples and $q(\cdot; \phi^{(t)})$ is trained on the generated sequences over $t$ training rounds.

- **DyNA PPO** (Angermueller et al., 2020): We closely follow the algorithm presented in (Angermueller et al., 2020). For a fair comparison, we use CNN ensembles to parameterize the proxy model instead of suggested architectures.

- **CMA-ES** (Hansen, 2006): We implement a covariance matrix adaptation evolution strategy (CMA-ES) for sequence generation. As the generated samples from CMA-ES are continuous, we convert the continuous vectors into one-hot representation by computing the argmax at each sequence position.

- **BO** (Snoek et al., 2012): We use classical GP-BO algorithm for all tasks. For Gaussian Process Regressor (GPR), we use a default setting from the `sklearn` library. For the acquisition function, they use Thompson sampling (Russo et al., 2018).

Furthermore, we employ GFN-AL and GFNSeqEditor. We adopt the original implementation and setup for TF-Bind-8 and AMP. For newly added tasks, we report better results among the original MLP policy and the LSTM policy. Note that GFP in FLEXS is different from the one employed in GFA-AL; we treat this as a new task based on the observation in the work from Surana et al. (2024).

- **GFN-AL** (Jain et al., 2022): We strictly follow hyperparameters of the original code in they conduct experiments on TF-Bind-8 and AMP. The proxy is parameterized using an MLP with two layers of 2,048 hidden. For the policy, a 2-layer MLP with 2,048 hidden dimensions is used, but we also test it with a 2-layer LSTM policy.

- **GFNSeqEditor** (Ghari et al., 2023): We implemented the editing procedure on top of the GFN-AL. Note that GFNSeqEditor does not utilize the proxy model, so the GFlowNets policy is trained using offline data only with the same policy training procedure of GFN-AL. GFNSeqEditor can also implicitly control the edit percentage with its hyperparameters, which are set $\delta = 0.01, \sigma = 0.0001, \lambda = 0.1$ in this study. Note that $\delta$ is not the conservativeness parameter.

# B FURTHER STUDIES

## B.1 STUDIES ON EFFECT OF $\delta$

### B.1.1 HARD TF-BIND-8

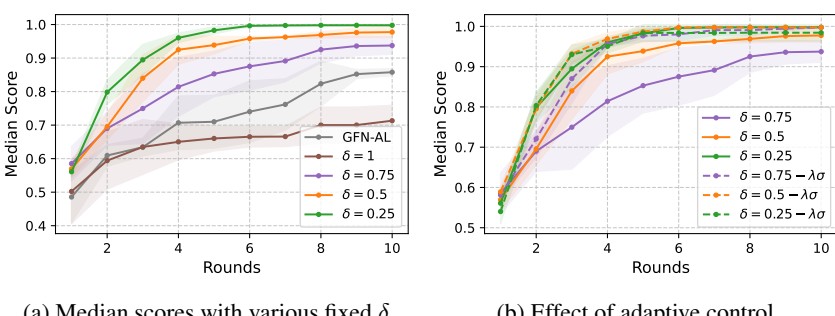

(a) Median scores with various fixed $\delta$      (b) Effect of adaptive control

Figure 5: Median score over rounds on Hard TF-Bind-8.

To verify its effectiveness and give intuition about how to set $\delta$, we conduct experiments with various $\delta$ in Hard TF-Bind-8. The results show that $\delta$-CS with $\delta < 1$ can significantly outperform GFN-AL by searching for data points that correlate better with the oracle. In the Hard TF-Bind-8 task, a more conservative search with $\delta = 0.25$ is beneficial since the proxy is unreliable in the early rounds. Note that the median scores with $\delta = 0.25$ and $0.5$ are higher than the median score of AdaLead, which is 0.928. In particular, ours with $\delta = 1$ means the full on-policy search (no conservativeness). The performance differences between $\delta = 1$ and GFN-AL came from $\epsilon$-noisy behavior policy, which selects random actions with a probability of 0.001, in GFN-AL. Furthermore, using adaptive $\delta(x, \sigma)$ mostly gives the improved scores as depicted in Figure 5b.

### B.1.2 RNA

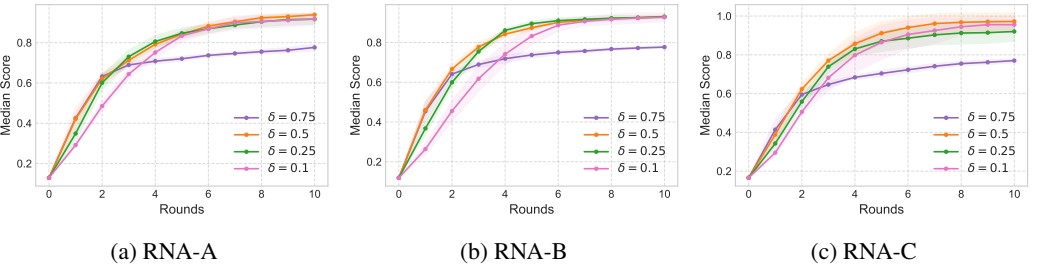

(a) RNA-A      (b) RNA-B      (c) RNA-C

Figure 6: Adaptive delta with various $\delta_{\text{const}}$ on RNA

### B.1.3 PROTEIN DESIGNS

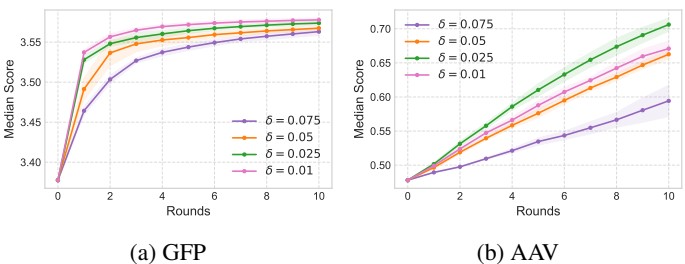

(a) GFP      (b) AAV

Figure 7: Adaptive delta with various $\delta_{\text{const}}$ on protein designs

## B.2    STUDIES ON THE EFFECT OF ADAPTIVE $\delta$ ON RNA

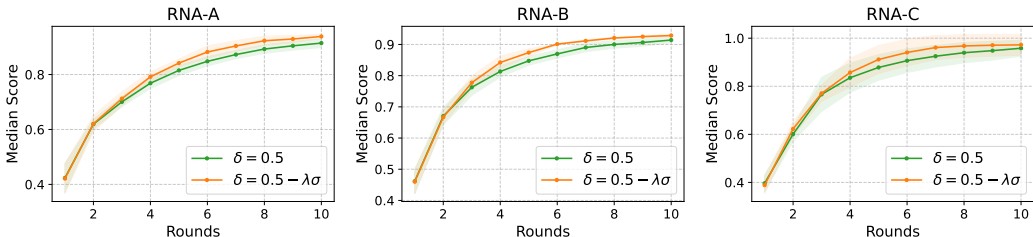

Figure 8: Effect of adaptive control on RNA

We examine the effects of proxy uncertainty-based $\delta$. In RNA, the average proxy standard deviation $\bar{\sigma}$ at the initial round is observed as 0.005 to 0.012. Therefore, we set $\lambda = 5$ to roughly make $\lambda\bar{\sigma} \approx 1/L$, where $L = 14$. As illustrated in Figure 8, $\delta(x;\sigma)$ consistently gives the higher score. However, the constant $\delta = 5$ still outperforms all baselines, exhibiting the robustness of $\delta$-CS.

### B.3 BALANCING CAPABILITY WITH VARIOUS $\delta$

Similar to Section 6.5, we also verify the balancing capability of $\delta$-CS on RNA-B and RNA-C. The $\delta$ is set from 0.1 to 0.5.

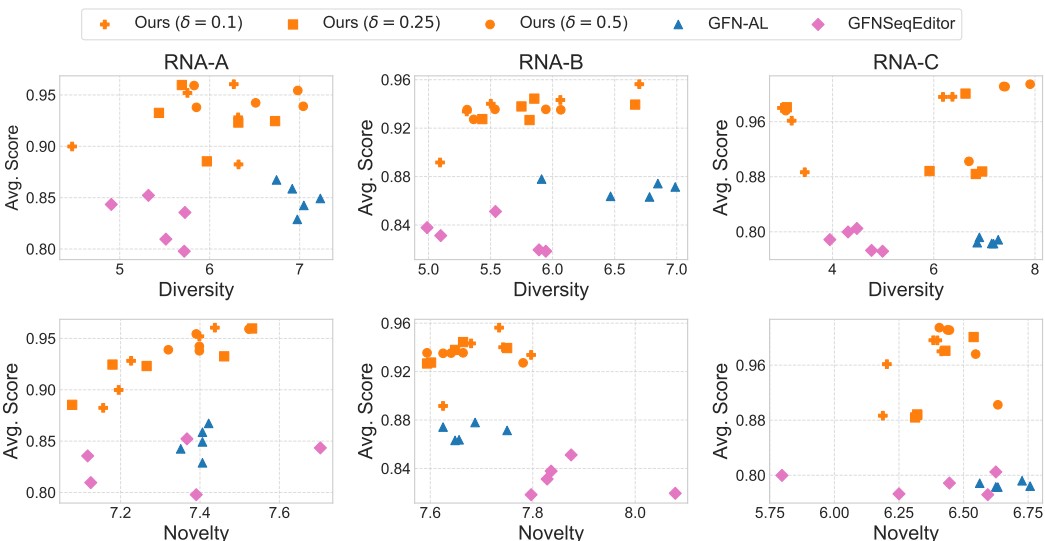

Figure 9: Average score and diversity/novelty with on RNA designs with various $\delta$.

For GFP and AAV, the $\delta$ is set from 0.01 to 0.05.

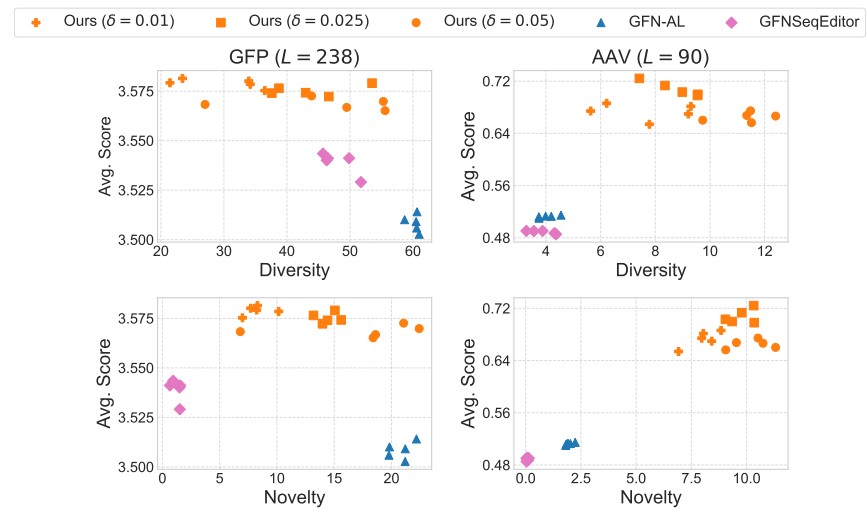

Figure 10: Average score and diversity/novelty on protein designs with various $\delta$.

### B.4 EXPERIMENTS WITH DIFFERENT PROXY ARCHITECTURE

We conducted additional experiments using different proxies in AAV, GFP, and RNA tasks with MuFacNet (Ren et al., 2022), whereas the existing proxy model is based on CNN architecture (a common benchmark). The results in Figures 11 and 12 show that the trends are consistent regardless of proxy models.

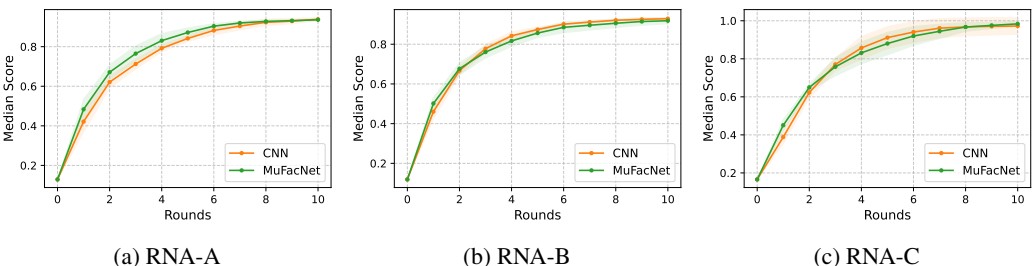

   (a) RNA-A               (b) RNA-B               (c) RNA-C

Figure 11: Comparison $\delta$-CS with CNN and MuFacNet on RNA

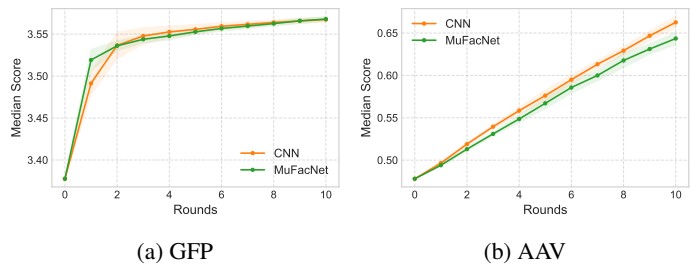

        (a) GFP              (b) AAV

Figure 12: Comparison $\delta$-CS with CNN and MuFacNet on protein designs

## B.5 COMPARISON WITH BACK-AND-FORTH SEARCH

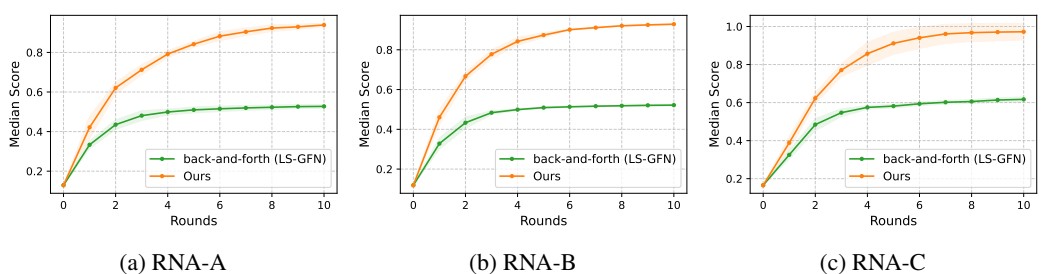

(a) RNA-A          (b) RNA-B          (c) RNA-C

Figure 13: Comparison $\delta$-CS with back-and-forth search in LS-GFN on RNA

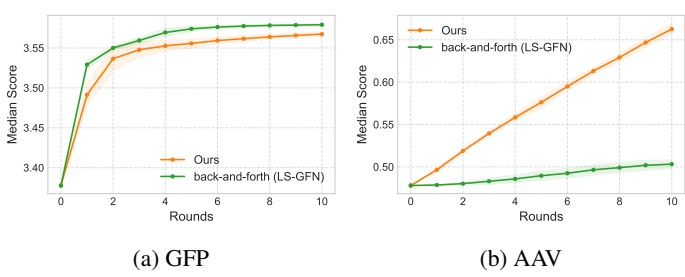

(a) GFP          (b) AAV

Figure 14: Comparison $\delta$-CS with back-and-forth search in LS-GFN on protein designs

### B.6 STUDIES ON QUERY BATCH SIZE PER ACTIVE ROUND

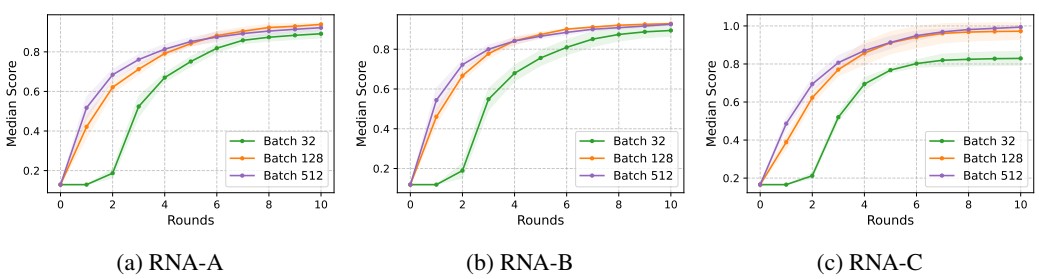

(a) RNA-A  (b) RNA-B  (c) RNA-C

Figure 15: Ablation studies of the number of queries (batch) per round on RNA

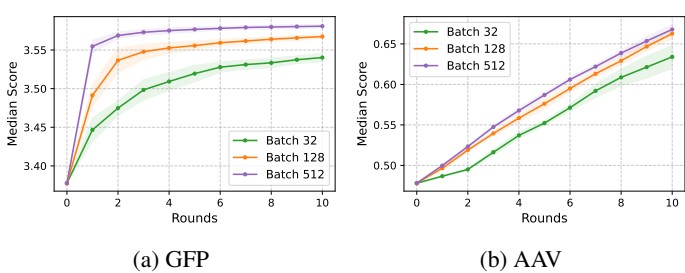

(a) GFP  (b) AAV

Figure 16: Ablation studies of the number of queries (batch) per round on protein designs

## B.7 COMPARISON WITH TURBO

We conducted experiments with TuRBO (Eriksson et al., 2019), a widely used trust region-based BO method for our setting. As shown in the following table, while TuRBO exhibits generally higher scores than classical BO, our method surpasses TuRBO across various tasks, exhibiting the superiority of our $\delta$-CS constraints.

Table 3: Maximum scores

|  | RNA-A ($L=14$) | RNA-B ($L=14$) | RNA-C ($L=14$) | TF-Bind-8 ($L=8$) | GFP ($L=238$) | AAV ($L=90$) |
|---|---|---|---|---|---|---|
| BO | $0.722 \pm 0.025$ | $0.720 \pm 0.032$ | $0.506 \pm 0.003$ | $0.977 \pm 0.008$ | $3.572 \pm 0.000$ | $0.500 \pm 0.000$ |
| TuRBO | $0.935 \pm 0.034$ | $0.921 \pm 0.052$ | $0.912 \pm 0.036$ | $0.974 \pm 0.019$ | $3.586 \pm 0.000$ | $0.500 \pm 0.000$ |
| Ours | $1.055 \pm 0.000$ | $1.014 \pm 0.001$ | $1.094 \pm 0.045$ | $0.981 \pm 0.002$ | $3.592 \pm 0.003$ | $0.708 \pm 0.010$ |

Table 4: Median scores

|  | RNA-A ($L=14$) | RNA-B ($L=14$) | RNA-C ($L=14$) | TF-Bind-8 ($L=8$) | GFP ($L=238$) | AAV ($L=90$) |
|---|---|---|---|---|---|---|
| BO | $0.510 \pm 0.008$ | $0.502 \pm 0.013$ | $0.506 \pm 0.003$ | $0.806 \pm 0.007$ | $3.378 \pm 0.000$ | $0.478 \pm 0.000$ |
| TuRBO | $0.622 \pm 0.046$ | $0.629 \pm 0.030$ | $0.541 \pm 0.068$ | $0.974 \pm 0.019$ | $3.583 \pm 0.003$ | $0.500 \pm 0.000$ |
| Ours | $0.939 \pm 0.008$ | $0.929 \pm 0.004$ | $0.972 \pm 0.043$ | $0.971 \pm 0.006$ | $3.567 \pm 0.003$ | $0.663 \pm 0.007$ |

## B.8 DIVERSITY AND NOVELTY

Following Jain et al. (2022), diversity and novelty are measured as follows.

$$\text{Diversity}(\mathcal{D}) = \frac{\sum_{(x_i, y_i) \in \mathcal{D}} \sum_{(x_j, y_j) \in \mathcal{D} \setminus \{(x_i, y_i)\}} d(x_i, x_j)}{|\mathcal{D}|(|\mathcal{D}| - 1)}$$

$$\text{Novelty}(\mathcal{D}) = \frac{\sum_{(x_i, y_i) \in \mathcal{D}} \min_{s_j \in \mathcal{D}_0} d(x_i, s_j)}{|\mathcal{D}|}$$

For a better comprehensive analysis, the diversity and novelty over rounds are illustrated in the following figures.

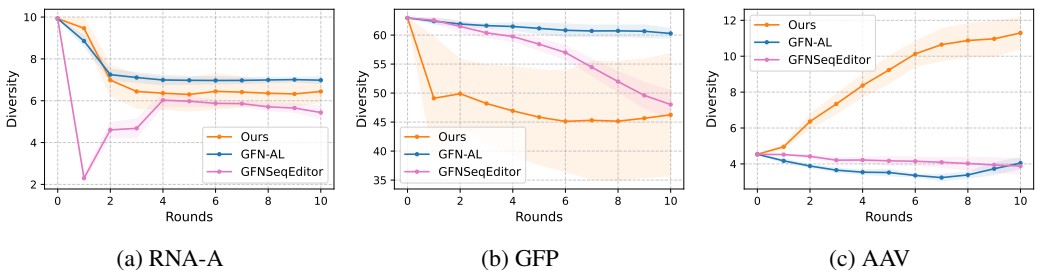

(a) RNA-A         (b) GFP         (c) AAV

Figure 17: Diversity over rounds

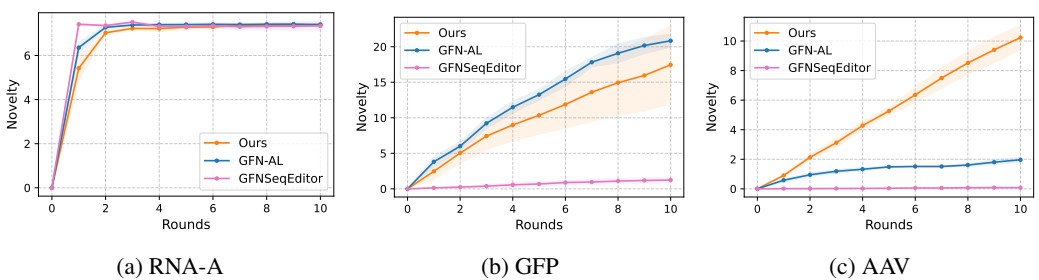

(a) RNA-A         (b) GFP         (c) AAV

Figure 18: Novelty over rounds

## B.9 Studies on Rank-Based Reweighted Sampling in Proxy Training

The rank-based reweighing also can be used in proxy training, i.e., $x \sim P_{\mathcal{D}_{t-1}}(x; k)$, where $k$ is a reweighting factor and fixed as 0.01 in this work. The results show that rank-based reweighted proxy training improves performance mostly. However, the gap is small, and $\delta$-CS still works well even without reweighting.

Table 5: Ablation studies of rank-based reweighted proxy training

|  |  | Max | Median | Mean | Diversity | Novelty |
|---|---|---|---|---|---|---|
| RNA-A | with rank-based | $1.055 \pm 0.000$ | $0.939 \pm 0.008$ | $0.947 \pm 0.009$ | $6.442 \pm 0.525$ | $7.406 \pm 0.066$ |
|  | without rank-based | $1.049 \pm 0.010$ | $0.936 \pm 0.016$ | $0.944 \pm 0.015$ | $5.782 \pm 0.697$ | $7.397 \pm 0.098$ |
| RNA-B | with rank-based | $1.014 \pm 0.001$ | $0.929 \pm 0.004$ | $0.934 \pm 0.003$ | $5.644 \pm 0.307$ | $7.661 \pm 0.064$ |
|  | without rank-based | $1.009 \pm 0.008$ | $0.932 \pm 0.012$ | $0.938 \pm 0.012$ | $6.252 \pm 0.291$ | $7.673 \pm 0.033$ |
| RNA-C | with rank-based | $1.094 \pm 0.045$ | $0.972 \pm 0.043$ | $0.983 \pm 0.043$ | $6.493 \pm 1.751$ | $6.494 \pm 0.084$ |
|  | without rank-based | $1.097 \pm 0.022$ | $0.958 \pm 0.029$ | $0.965 \pm 0.031$ | $5.472 \pm 1.921$ | $6.464 \pm 0.192$ |

## B.10 Ablation Studies on Different Initial Dataset Sizes

In this section, we extend our ablation study by comparing our method, $\delta$-CS, against two baselines: GFN-AL and an additional off-policy search method, LS-GFN (Kim et al., 2024d). LS-GFN incorporates a back-and-forth search strategy that partially backtracks trajectories using a backward policy and reconstructs them using a forward policy of the GFN. This study evaluates the performance of $\delta$-CS under varying initial dataset sizes, $|\mathcal{D}_0| = 1,000$, and compares the results to the original ablation study setup.

As shown in Table 6, our $\delta$-CS demonstrates a substantial advantage over both GFN-AL and its improved variant, LS-GFN, which leverages back-and-forth search. The results highlight the effectiveness of $\delta$-CS in enhancing GFN training by enabling a more robust and conservative off-policy search, which is critical for improving proxy-based active learning.

Table 6: Ablation study results with 1,000 initial datapoints for GFP and AAV tasks, showing maximum values achieved after active learning.

| Method | GFP | AAV |
|---|---|---|
| Adalead | $3.568 \pm 0.005$ | $0.557 \pm 0.023$ |
| GFN-AL | $3.586 \pm 0.006$ | $0.560 \pm 0.008$ |
| GFN-AL + LS (Kim et al., 2024d) | $3.580 \pm 0.003$ | $0.493 \pm 0.006$ |
| **GFN-AL + $\delta$-CS** | $\mathbf{3.591 \pm 0.007}$ | $\mathbf{0.704 \pm 0.024}$ |

## B.11    ABLATION STUDIES ON VARIOUS PROXY MODEL QUALITIES

To further verify that $\delta$-CS is robust to proxy misspecification compared to other GFN methods, we conducted additional experiments to test whether this hypothesis holds at different levels of proxy model quality.

To degrade the proxy model quality, we truncated the initial dataset at different levels—50%, 25%, and 10% percentiles based on reward values. Proxy models trained on datasets with lower percentile cutoffs are more misspecified for higher-reward data points, making GFN training and search more challenging. Under these circumstances, we compared our method with GFN-AL and LS-GFN (Kim et al., 2024d) as GFN baselines.

As shown in the figure above, the performance decreases as the percentile decreases, which is expected because the proxy quality deteriorates significantly. Among the baselines, our method consistently provides substantially better performance than the others. This demonstrates that our hypothesis—that a conservative search with $\delta$-CS is necessary—holds across different levels of proxy model quality.

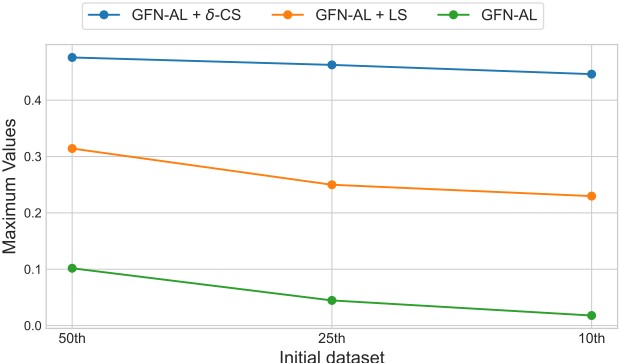

Figure 19: Maximum values achieved after active learning with varying initial dataset quality (AAV task).

## C  FULL RESULTS OF MAIN RESULTS

### C.1  FULL RESULTS OF RNA SEQUENCE DESIGN

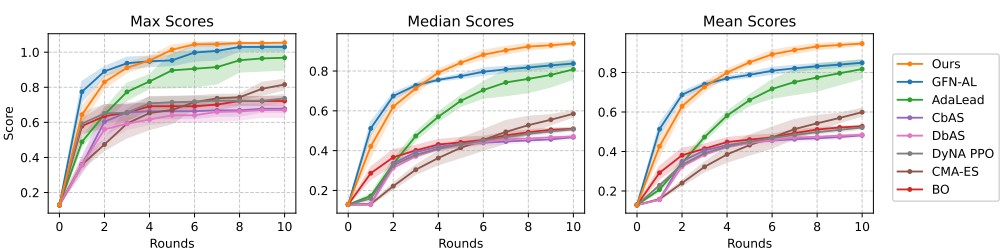

Figure 20: The max, median, and mean curve over rounds in RNA-A

Table 7: The results of RNA-A after ten rounds.

|  | Max | Median | Mean | Diversity | Novelty |
|---|---|---|---|---|---|
| AdaLead | $0.968 \pm 0.070$ | $0.808 \pm 0.049$ | $0.817 \pm 0.048$ | $3.518 \pm 0.446$ | $6.888 \pm 0.426$ |
| BO | $0.722 \pm 0.025$ | $0.510 \pm 0.008$ | $0.528 \pm 0.004$ | $\mathbf{9.531 \pm 0.062}$ | $5.842 \pm 0.083$ |
| CMA-ES | $0.816 \pm 0.030$ | $0.585 \pm 0.016$ | $0.599 \pm 0.020$ | $5.747 \pm 0.110$ | $6.373 \pm 0.159$ |
| CbAS | $0.678 \pm 0.020$ | $0.467 \pm 0.009$ | $0.481 \pm 0.008$ | $9.457 \pm 0.189$ | $5.428 \pm 0.078$ |
| DbAS | $0.670 \pm 0.041$ | $0.472 \pm 0.016$ | $0.485 \pm 0.015$ | $9.483 \pm 0.100$ | $5.450 \pm 0.132$ |
| DyNA PPO | $0.737 \pm 0.022$ | $0.507 \pm 0.007$ | $0.521 \pm 0.009$ | $8.889 \pm 0.034$ | $5.828 \pm 0.095$ |
| GFN-AL | $1.030 \pm 0.024$ | $0.838 \pm 0.013$ | $0.849 \pm 0.013$ | $6.983 \pm 0.159$ | $7.398 \pm 0.024$ |
| GFN-AL + $\delta$-CS | $\mathbf{1.055 \pm 0.000}$ | $\mathbf{0.939 \pm 0.008}$ | $\mathbf{0.947 \pm 0.009}$ | $6.442 \pm 0.525$ | $\mathbf{7.406 \pm 0.066}$ |

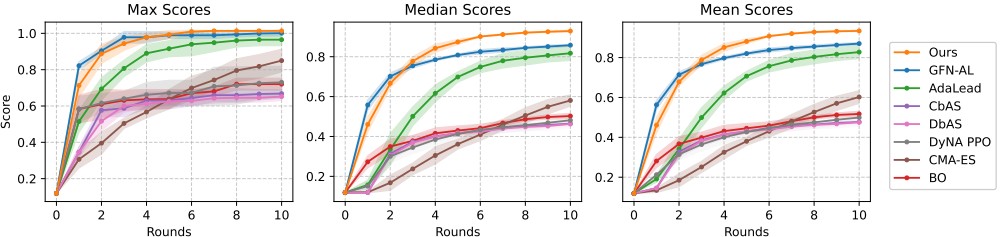

Figure 21: The max, median, and mean curve over rounds in RNA-B

Table 8: The results of RNA-B after ten rounds.

|  | Max | Median | Mean | Diversity | Novelty |
|---|---|---|---|---|---|
| AdaLead | $0.965 \pm 0.033$ | $0.817 \pm 0.036$ | $0.828 \pm 0.032$ | $3.334 \pm 0.423$ | $7.441 \pm 0.135$ |
| BO | $0.720 \pm 0.032$ | $0.502 \pm 0.013$ | $0.517 \pm 0.014$ | $\mathbf{9.495 \pm 0.103}$ | $5.903 \pm 0.116$ |
| CMA-ES | $0.850 \pm 0.063$ | $0.581 \pm 0.028$ | $0.602 \pm 0.032$ | $5.568 \pm 0.365$ | $6.480 \pm 0.200$ |
| CbAS | $0.668 \pm 0.021$ | $0.465 \pm 0.005$ | $0.477 \pm 0.004$ | $9.234 \pm 0.356$ | $5.523 \pm 0.083$ |
| DbAS | $0.652 \pm 0.021$ | $0.463 \pm 0.019$ | $0.475 \pm 0.019$ | $9.019 \pm 0.648$ | $5.537 \pm 0.150$ |
| DyNA PPO | $0.730 \pm 0.088$ | $0.481 \pm 0.028$ | $0.499 \pm 0.029$ | $8.978 \pm 0.196$ | $5.839 \pm 0.198$ |
| GFN-AL | $1.001 \pm 0.016$ | $0.858 \pm 0.004$ | $0.870 \pm 0.006$ | $6.599 \pm 0.384$ | $\mathbf{7.673 \pm 0.043}$ |
| GFN-AL + $\delta$-CS | $\mathbf{1.014 \pm 0.001}$ | $\mathbf{0.929 \pm 0.004}$ | $\mathbf{0.934 \pm 0.003}$ | $5.644 \pm 0.307$ | $7.661 \pm 0.064$ |

### C.2  FULL RESULTS OF TF-BIND-8

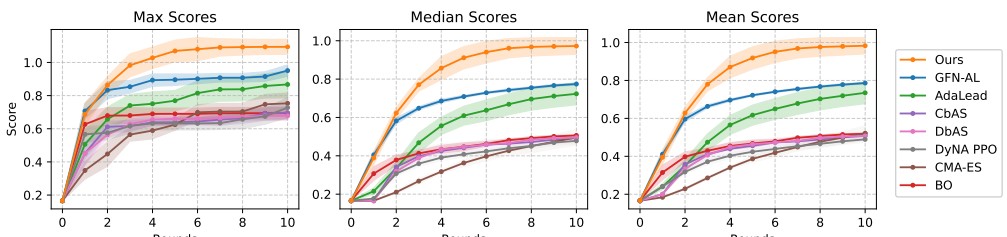

Figure 22: The max, median, and mean curve over rounds in RNA-C

Table 9: The results of RNA-C after ten rounds.

|  | Max | Median | Mean | Diversity | Novelty |
|---|---|---|---|---|---|
| AdaLead | $0.867 \pm 0.081$ | $0.723 \pm 0.057$ | $0.735 \pm 0.057$ | $3.893 \pm 0.444$ | $5.856 \pm 0.515$ |
| BO | $0.694 \pm 0.034$ | $0.506 \pm 0.003$ | $0.519 \pm 0.003$ | $9.714 \pm 0.054$ | $5.430 \pm 0.043$ |
| CMA-ES | $0.753 \pm 0.062$ | $0.496 \pm 0.041$ | $0.521 \pm 0.037$ | $5.581 \pm 0.399$ | $5.019 \pm 0.294$ |
| CbAS | $0.696 \pm 0.041$ | $0.492 \pm 0.018$ | $0.507 \pm 0.017$ | $\mathbf{9.518 \pm 0.310}$ | $5.033 \pm 0.086$ |
| DbAS | $0.678 \pm 0.025$ | $0.495 \pm 0.010$ | $0.508 \pm 0.011$ | $9.249 \pm 0.414$ | $5.128 \pm 0.153$ |
| DyNA PPO | $0.728 \pm 0.060$ | $0.478 \pm 0.015$ | $0.489 \pm 0.015$ | $9.246 \pm 0.086$ | $5.306 \pm 0.124$ |
| GNF-AL | $0.951 \pm 0.034$ | $0.774 \pm 0.004$ | $0.786 \pm 0.004$ | $7.072 \pm 0.163$ | $\mathbf{6.661 \pm 0.071}$ |
| GFN-AL + $\delta$-CS | $\mathbf{1.094 \pm 0.045}$ | $\mathbf{0.972 \pm 0.043}$ | $\mathbf{0.983 \pm 0.043}$ | $6.493 \pm 1.751$ | $6.494 \pm 0.084$ |

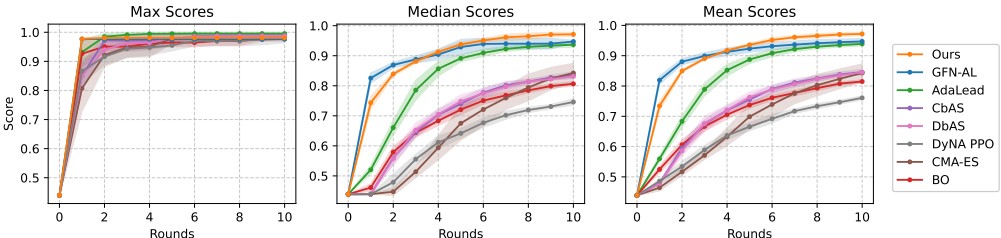

Figure 23: The max, median, and mean curve over rounds in TF-Bind-8

Table 10: The results of TF-Bind-8 after ten rounds.

|  | Max | Median | Mean | Diversity | Novelty |
|---|---|---|---|---|---|
| AdaLead | $\mathbf{0.995 \pm 0.004}$ | $0.937 \pm 0.008$ | $0.939 \pm 0.007$ | $3.506 \pm 0.267$ | $1.194 \pm 0.035$ |
| BO | $0.977 \pm 0.008$ | $0.806 \pm 0.007$ | $0.815 \pm 0.005$ | $4.824 \pm 0.074$ | $1.144 \pm 0.029$ |
| CMA-ES | $0.986 \pm 0.008$ | $0.843 \pm 0.032$ | $0.843 \pm 0.030$ | $3.617 \pm 0.321$ | $1.130 \pm 0.083$ |
| CbAS | $0.988 \pm 0.004$ | $0.835 \pm 0.011$ | $0.845 \pm 0.009$ | $4.662 \pm 0.079$ | $1.134 \pm 0.021$ |
| DbAS | $0.987 \pm 0.004$ | $0.831 \pm 0.005$ | $0.845 \pm 0.005$ | $4.694 \pm 0.056$ | $1.141 \pm 0.047$ |
| DyNA PPO | $0.977 \pm 0.013$ | $0.746 \pm 0.010$ | $0.761 \pm 0.006$ | $4.430 \pm 0.030$ | $1.120 \pm 0.021$ |
| GFN-AL | $0.976 \pm 0.002$ | $0.947 \pm 0.004$ | $0.947 \pm 0.009$ | $3.158 \pm 0.166$ | $\mathbf{2.409 \pm 0.071}$ |
| GFN-AL + $\delta$-CS | $0.981 \pm 0.002$ | $\mathbf{0.971 \pm 0.006}$ | $\mathbf{0.972 \pm 0.005}$ | $\mathbf{1.277 \pm 0.182}$ | $2.237 \pm 0.356$ |

## C.3 Full results of Protein design

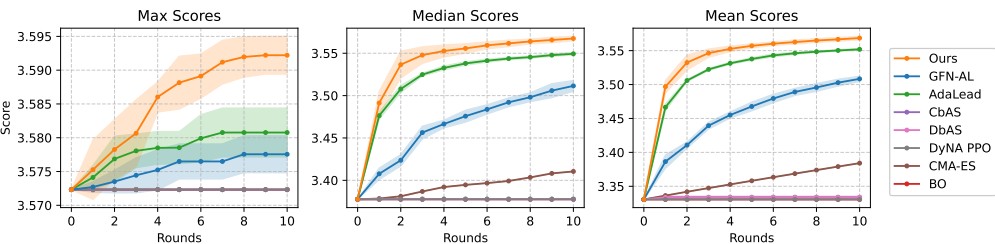

Figure 24: The max, median, and mean curve over rounds in GFP

Table 11: The results of GFP after ten rounds.

|  | Max | Median | Mean | Diversity | Novelty |
|---|---|---|---|---|---|
| AdaLead | $3.581 \pm 0.004$ | $3.549 \pm 0.002$ | $3.552 \pm 0.002$ | $47.237 \pm 1.213$ | $1.467 \pm 0.094$ |
| BO | $3.572 \pm 0.000$ | $3.378 \pm 0.000$ | $3.331 \pm 0.000$ | $\mathbf{62.955 \pm 0.000}$ | $0.000 \pm 0.000$ |
| CMA-ES | $3.572 \pm 0.000$ | $3.410 \pm 0.000$ | $3.384 \pm 0.000$ | $58.299 \pm 0.000$ | $0.000 \pm 0.000$ |
| CbAS | $3.572 \pm 0.000$ | $3.378 \pm 0.000$ | $3.334 \pm 0.002$ | $62.926 \pm 0.139$ | $0.009 \pm 0.012$ |
| DbAS | $3.572 \pm 0.000$ | $3.378 \pm 0.000$ | $3.334 \pm 0.002$ | $62.926 \pm 0.139$ | $0.009 \pm 0.012$ |
| DyNA PPO | $3.572 \pm 0.000$ | $3.378 \pm 0.000$ | $3.331 \pm 0.000$ | $\mathbf{62.955 \pm 0.000}$ | $0.000 \pm 0.000$ |
| GFN-AL | $3.578 \pm 0.003$ | $3.511 \pm 0.006$ | $3.508 \pm 0.004$ | $60.278 \pm 0.819$ | $\mathbf{20.837 \pm 0.916}$ |
| GFN-AL + $\delta$-CS | $\mathbf{3.592 \pm 0.003}$ | $\mathbf{3.567 \pm 0.003}$ | $\mathbf{3.569 \pm 0.003}$ | $46.255 \pm 10.534$ | $17.459 \pm 5.538$ |

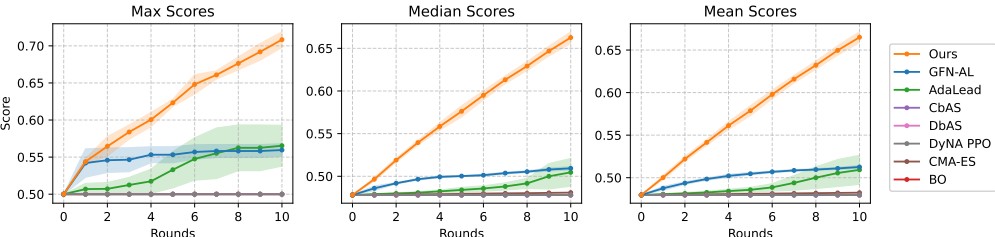

Figure 25: The max, median, and mean curve over rounds in AAV

Table 12: The results of AAV after ten rounds.

|  | Max | Median | Mean | Diversity | Novelty |
|---|---|---|---|---|---|
| AdaLead | $0.565 \pm 0.027$ | $0.505 \pm 0.016$ | $0.509 \pm 0.017$ | $5.693 \pm 0.946$ | $2.133 \pm 1.266$ |
| BO | $0.500 \pm 0.000$ | $0.478 \pm 0.000$ | $0.480 \pm 0.000$ | $4.536 \pm 0.000$ | $0.000 \pm 0.000$ |
| CMA-ES | $0.500 \pm 0.000$ | $0.481 \pm 0.000$ | $0.482 \pm 0.000$ | $4.148 \pm 0.000$ | $0.000 \pm 0.000$ |
| CbAS | $0.500 \pm 0.000$ | $0.478 \pm 0.000$ | $0.480 \pm 0.000$ | $4.545 \pm 0.018$ | $0.002 \pm 0.003$ |
| DbAS | $0.500 \pm 0.000$ | $0.478 \pm 0.000$ | $0.480 \pm 0.000$ | $4.545 \pm 0.018$ | $0.002 \pm 0.003$ |
| DyNA PPO | $0.500 \pm 0.000$ | $0.478 \pm 0.000$ | $0.480 \pm 0.000$ | $4.536 \pm 0.000$ | $0.000 \pm 0.000$ |
| GFN-AL | $0.560 \pm 0.008$ | $0.509 \pm 0.002$ | $0.513 \pm 0.002$ | $4.044 \pm 0.303$ | $1.966 \pm 0.157$ |
| GFN-AL + $\delta$-CS | $\mathbf{0.708 \pm 0.010}$ | $\mathbf{0.663 \pm 0.007}$ | $\mathbf{0.665 \pm 0.006}$ | $\mathbf{11.296 \pm 0.865}$ | $\mathbf{10.233 \pm 0.822}$ |

