# OpenReview forum: "Improved Off-policy Reinforcement Learning in Biological Sequence Design"
_ICLR.cc/2025/Conference — Submitted to ICLR 2025_

### Official Review · Reviewer_6RU1 · 2024-10-22

**Soundness:** 3
**Presentation:** 2
**Contribution:** 2
**Rating:** 5
**Confidence:** 3

**Summary:**

The authors propose an off-policy search method, δ-Conservative Search (δ-CS), a novel off-policy search method for training GFlowNets designed to improve robustness against proxy misspecification. The proxy misspecification is critical in the biological sequence design problem due to its vastness of the search space and high cost of evaluating candidate sequences.

The key of the δ-CS method is its dynamic control of search conservativeness through a parameter δ, which balances the trade-off between exploring new sequences and staying within the reliable regions of the proxy model. The method injects noise into high-scoring offline sequences by masking tokens randomly and then denoises them using a GFlowNet policy. This approach ensures that the search process is guided by the uncertainty of the proxy model, making it robust against proxy mis-specification, particularly for out-of-distribution sequences.

In summary, δ-CS offers a scalable, adaptive, and effective framework for generating novel and high-reward biological sequences, providing valuable contributions to the fields of synthetic biology and biotechnology.

**Strengths:**

**Novel Contribution**: A key challenge in previous active learning framework is that, the proxy model often produces
highly unreliable rewards $R(x; \phi)$ for out-of-distribution inputs. This paper mitigates this by providing off-policy trajectories denoised from noise-injected high-reward sequences. This paper uses $\delta$ to control the how "novel" the denoised sequences are compared with existing ones. For example, $\delta=1$ means the full on-policy search.

**Comprehensive Experiments**: The experiments include a range of biological sequence design tasks, including DNA, RNA, protein, and peptide design. The comparison with several baselines, including DyNA PPO and other Bayesian optimization methods, further strengthens the claims.

**Weaknesses:**

**Complexity in Implementation**: Although the proposed method shows strong performance, its implementation appears to be complex. The exploration in the active learning process is controlled by $\delta$, making its adaptation crucial. The current adaptation is controlled by the hyperparameters $\lambda, \delta_{\rm const}$. Additionally, the selection of high-score offline sequences is based on a rank-based reweighted prior, with hyperparameter ($k$). There are also potential alternatives for selecting high-reward offline sequences and adapting $\delta$. As a result, it remains unclear whether the "three steps" outlined in Lines 187-189 offer a general improvement over standard active learning methods or if the paper has cherry-picked one favorable combination.

**Uncertainty Estimate**: The method assumes that the proxy model can reliably estimate uncertainty. However, as noted in the paper, the proxy model is unreliable for out-of-distribution (OOD) inputs. This raises a concern: while the paper does not trust the proxy's output for OOD inputs during search, it still relies on the proxy to compute uncertainty for all inputs. This assumption seems questionable. I know that the experiments in Appendix B demonstrate that this adaptation is effective, but I want to hear the opinions from the authors on that.


**Choice of GFlowNets** The paper employs GFlowNets for the policy, which are equivalent to soft off-policy reinforcement learning algorithms. However, it does not explore whether the $\delta$-conservative search would be effective with other search methods. Furthermore, in Tables 1 and 2, the performance gain of GFN-AL + $\delta$-C over GFN-AL does not appear to be substantial compared to the performance gains of GFN-AL itself. This suggests that the main improvement might stem from the use of GFlowNets rather than from the search method. Many studies (to name a few, [1,2,3,4,5]) claim significant improvements over the standard GFlowNet (used in this paper), so it might be worth investigating whether replacing the GFlowNet in GFN-AL with a modified version could lead to even better performance than modifying the search.

[1] Local Search GFlowNets. Minsu Kim, et al. ICLR 2024 spotlight.
[2] Learning Energy Decompositions for Partial Inference of GFlowNets. Hyosoon Jang, et al. ICLR 2024 oral.
[3] Order-Preserving GFlowNets. Yihang Chen, et al. ICLR 2024.
[4] Learning to Scale Logits for Temperature-Conditional GFlowNets. Minsu Kim, et al. ICML 2024.
[5] QGFN: Controllable Greediness with Action Values. Elaine Lau, et al. NeurIPS 2024.

**Questions:**

See weaknesses.

---

> ### Author Response · Authors · 2024-11-22
> **Response (1/2)**
>
> ### W1: Complexity in implementation
>
> We want to emphasize that the hyperparameters are not excessively tuned. First, the reweighting coefficient $k$ is usually set to 0.01 or 0.001 [1, 2]; the results below (and more in Appendix B.8) show that both settings outperform the baselines. The adaptation is not critical for $\delta$-CS, as performance is decent with just a constant hyperparameter and is robust to its selection. $\lambda$ (which is only required for the adaptive version of $\delta$-CS) is also easy to tune because it is just a scaling factor. We can use a tuned $\lambda$ for similar scale tasks (e.g., we use identical $\lambda$ for short-scale tasks like DNA and RNA, and long-scale tasks like AAV and GFP), which can be adjusted with a simple validation experiment.
>
> To demonstrate hyperparameter robustness, our sensitivity analysis results in Appendix B.2 and the following tables show that our method performs well with various values of these parameters, highlighting its robustness.
>
> [1] Kim, Minsu, et al. "Bootstrapped training of score-conditioned generator for offline design of biological sequences." NeurIPS. 2023.
> [2] Kim, Hyeonah, et al. "Genetic-guided GFlowNets for sample efficient molecular optimization." NeurIPS. 2024.
>
> **Ablation for $k$**
> |         | $k=0.01$ (default) | $k=0.001$ |
> | - | - | - |
> | RNA-A (Max) | 1.055 ± 0.000 | 1.036 ± 0.023 |
> | GFP (Max)   | 3.592 ± 0.003 | 3.598 ± 0.003 |
>
> **RNA-A**
> |  | $\delta = 0.1$ | $\delta = 0.25$ | $\delta = 0.5$ | $\delta = 0.75$ | GFN-AL |
> | - | - | - | - | - | - |
> | Max | 1.040 ± 0.014 | 1.031 ± 0.026 | **1.055** ± 0.000 | 0.925 ± 0.055 | 1.030 ± 0.024 |
> | Median | 0.918 ± 0.034 | 0.916 ± 0.028 | **0.939** ± 0.008 | 0.776 ± 0.018 | 0.838± 0.013 |
>
> **GFP**
> | | $\delta = 0.01$ | $\delta = 0.025$ | $\delta = 0.05$ | $\delta = 0.075$ | GFN-AL |
> | - | - | - | - | - | - |
> | Max | 3.593 ± 0.003 | **3.594** ± 0.005 | 3.592 ± 0.003 | 3.592 ± 0.005 | 3.578 ± 0.003 |
> | Median | 	**3.578** ± 0.002 | 3.574 ± 0.003 | 3.567 ± 0.003 | 3.563 ± 0.004 | 3.511 ± 0.006 |
>
>
> It is noteworthy that we have not excessively searched these parameters, and the results are not cherry-picked. As evidences, the better combinations for each task are found in our additional results above e.g., $\delta=0.025$ or $k=0.001$ for GFP.
>
> ---
>
> ### W2: Uncertainty Estimate
>
> Good point. Theoretically, uncertainty can be measured even when prediction accuracy is low, as they have different characteristics. The low prediction accuracy of the proxy arises from **out-of-distribution problems**—data points not seen during training. Uncertainty is measured based on how such a new data point affects the uncertainty of the proxy function. This can be quantified through posterior inference over the proxy model's parameters, given the current **in-distribution training dataset**. Although theoretically valid, such posterior inference is difficult in practice. We use ensemble methods to estimate uncertainty, which are sub-optimal but still practical and widely used. Developing better Bayesian posterior inference methods for active learning could be valuable future work. Our method is orthogonal to this, as $\delta$-CS can be used with such improved uncertainty estimation methods.

---

> ### Author Response · Authors · 2024-11-22
> **Response (2/2)**
>
> ### W3: Choice of GFlowNets
>
>
> 1. **Performance gain of GFN-AL + $\delta$-CS over GFN-AL.**
>
> We argue that the performance improvement achieved by $\delta$-CS is significant, particularly evident in the large margins observed for RNA-C, GFP, and AAV. Notably, as shown in Table 2, while GFN-AL underperforms in terms of average score compared to the RL baseline (DyNA PPO), our approach (with EI) outperforms it in *all metrics*, mean and max score, diversity, and novelty.
>
> 2. **About [1,4,5]**
>
> We agree with this suggestion. Such off-policy advancements in the GFlowNet community can potentially improve active learning performance. While prior methods design off-policy exploration [1,4,5] under the assumption that the reward model is accurate, our delta-CS method is built on a different assumption and purpose: conducting conservative search to mitigate the risks from proxy misspecification. Among these prior methods, local search GFlowNets (LS-GFN) [1] seem to be very related to our method and can also be applied in conservative search. We directly compare our method with LS-GFN as follows:
>
> Results (MAXIMIUM)
>
> | | RNA-A (L=14)  | RNA-B (L=14)  | RNA-C (L=14)  | TFBind8 (L=8) | GFP (L=238)   | AAV (L=90)    |
> | - | - | - | - | - | - | - |
> | back-and-forth (LS-GFN) | 0.613 ± 0.009 | 0.572 ± 0.003 | 0.722 ± 0.009 | 0.977 ± 0.008 |  3.592 ± 0.001 | 0.549 ± 0.005 |
> | Ours  | 1.055 ± 0.000 | 1.014 ± 0.001 | 1.094 ± 0.045 | 0.981 ± 0.002 | 3.592 ± 0.003 | 0.708 ± 0.010 |
>
> As shown in the table, our method clearly outperforms LS-GFN. We suspect this is because the search flexibility of delta-CS is better than that of LS-GFN. The algorithm of LS-GFN is based on back-and-forth local search, where they use a backward policy to partially destroy a solution and a forward policy to reconstruct a new solution. However, in the auto-regressive sequence generation setting, such a backward policy must be unidirectional; therefore, their local search region is bounded in the leaf space of the sequences. On the other hand, delta-CS randomly destroys tokens independently, regardless of whether the token is located at the head or leaf node, allowing it to search more flexibly in the sequence space.
>
> We have included experimental results in Appendix B.5 and have added prior training methods of GFlowNets to the related works section.
>
> 3. **About [2,3]**
>
> Regarding other related works, references [2] and [3] have clearly different purposes and are **orthogonal** to ours, as they focus on improving the training scheme of GFlowNet but we focus to improve exploration which provides experiences for training. Specifically, [2] suggests a new loss function for better credit assignment, while [3] proposes a new method for multi-objective settings. We may utilize [2] with $\delta$-CS when the trajectory length is too large and better credit assignment is needed. Similarly, we might employ [3] with delta-CS when multi-objective active learning is required. These could be exciting directions for future work. We have added this to the discussion section of our manuscript.
>
> Thank you for pointing this out. We can greatly improve our manuscript.
>
> [1] Local Search GFlowNets. Minsu Kim, et al. ICLR 2024 spotlight.
>
> [2] Learning Energy Decompositions for Partial Inference of GFlowNets. Hyosoon Jang, et al. ICLR 2024 oral.
>
> [3] Order-Preserving GFlowNets. Yihang Chen, et al. ICLR 2024.
>
> [4] Learning to Scale Logits for Temperature-Conditional GFlowNets. Minsu Kim, et al. ICML 2024.
>
> [5] QGFN: Controllable Greediness with Action Values. Elaine Lau, et al. NeurIPS 2024.

---

> > ### Comment · Reviewer_6RU1 · 2024-11-23
> >
> > Thanks for your comments. Regarding the uncertainty estimate, I will keep my scores.

---

> ### Author Response · Authors · 2024-11-25
>
> Thanks for the feedback. We are pleased to know that your concerns except for regarding uncertainty have been addressed.
>
> We would like to emphasize that considering uncertainty in $\delta$ represents an additional component—the adaptive version of $\delta$. Furthermore, in the adaptive $\delta$, the uncertainty is measured for observed data sets, where the proxy is relatively reliable compared to OOD data points, with the properly chosen sailing factor $\lambda$. Our main message is that employing a conservative search using $\delta$ itself is beneficial for active learning in off-policy reinforcement learning for biological sequence design, as it effectively limits the search space of the generative policy. The subsequent experimental results demonstrate that even a constant $\delta$ without uncertainty measuring gives competitive results, outperforming other baselines.
>
> **maximum scores**
> |         | RNA-A | RNA-B | RNA-C |
> | ------- | -- | --- | - |
> | AdaLead | 0.968 ± 0.070 | 0.965 ± 0.033 | 0.867 ± 0.081 |
> | GFN-AL | 1.030 ± 0.024 | 1.001 ± 0.016 | 0.951 ± 0.034 |
> | Ours (Adaptive)   | 1.055 ± 0.000 | 1.014 ± 0.001 | 1.094 ± 0.045 |
> | Ours (Constant)   | 1.041 ± 0.023 | 1.014 ± 0.001 | 1.102 ± 0.024 |
>
>
>
> **median scores**
> |         | RNA-A | RNA-B | RNA-C |
> | ------- | -- | --- | - |
> | AdaLead |0.808 ± 0.049 | 0.817 ± 0.036 | 0.723 ± 0.057 |
> | GFN-AL | 0.838 ± 0.013 | 0.858 ± 0.004 | 0.774 ± 0.004 |
> | Ours (Adaptive)   | 0.939 ± 0.008 | 0.929 ± 0.004 | 0.972 ± 0.043 |
> | Ours (Constant)   | 0.914 ± 0.016 | 0.914 ± 0.009 | 0.958 ± 0.033 |

---

> > ### Author Response · Authors · 2024-11-26
> >
> > Dear Reviewer 6RU1,
> >
> > As we approach the end of the discussion period, we would like to ask if there is any additional information we can provide to address your evaluation of our paper. We believe our previous responses have effectively addressed your initial concerns and clarified key points that may have been unclear.
> >
> > In particular, we would like to confirm whether our responses regarding your concerns on uncertainty estimation were satisfactory. If not, could you please specify which aspects remain unresolved? To summarize, we aimed to clarify the following:
> >
> > 1. **Mean estimation vs. uncertainty estimation:** Mean estimation (a measure for new inputs) and uncertainty estimation (a measure of the prediction model itself) exhibit clearly different characteristics in out-of-distribution scenarios. Proxy misspecification pertains to the proxy's error in estimating the mean.
> >
> > 2. **High performance without accurate uncertainty estimation:** Our method's high performance does not rely on the accuracy of uncertainty estimation. As shown in our experiments, our performance is clearly superior to the baselines even without uncertainty-aware adaptive adjustments of $\delta$.
> >
> > **We kindly ask you to confirm if our responses address your concerns regarding uncertainty estimation, or if further clarification is needed.**
> >
> > Thank you once again for your thoughtful consideration of our work.
> >
> > Sincerely,
> > The Authors

---

> > > ### Author Response · Authors · 2024-12-02
> > >
> > > Dear Reviewer 6RU1,
> > >
> > > This is a gentle reminder that we are nearing the end of the discussion period. Please let us know if you have any additional concerns regarding uncertainty estimation. If our previous responses have resolved your concerns, we kindly ask you to consider adjusting your score accordingly.
> > >
> > > Best regards,
> > >
> > > The Authors

---

### Official Review · Reviewer_WGsn · 2024-11-02

**Soundness:** 2
**Presentation:** 3
**Contribution:** 2
**Rating:** 3
**Confidence:** 4

**Summary:**

On-policy reinforcement learning (RL) methods for biological sequence design tasks exhibit limited search flexibility. In contrast, off-policy methods like GFlowNets provide diversity-seeking capabilities and flexible exploration strategies. However, GFlowNets struggle with long sequence tasks due to the (hypothesis (*)) poor quality of the proxy model which serves as a reward function for training the policy network.
This paper addresses the problem of GFlowNets' ineffectiveness in long biological sequence design tasks due to the proxy model misspecifications for out-of-distribution inputs and proposes $\delta$-CS, an off-policy search method for training GFlowNets to enhance robustness.
$\delta$-CS addresses this issue by incorporating a conservativeness parameter, $\delta$, which can adaptively control the search boundaries. This approach effectively restricts the search to more reliable regions, mitigating the impact of inaccurate proxy models for out-of-distribution data points.

**Strengths:**

- $\delta$-CS provides a balance between novelty and conservativeness.
- An adaptive variant of $\delta$-CS adjusts the level of conservativeness dynamically based on the difficulty of the search landscape, which may vary in practical applications.
- The paper provides empirical evidence by validating $\delta$-CS across various biological sequence design tasks (DNA, RNA, protein, peptide) against baseline methods.
- The paper is generally well-written, although there are some minor descriptive confusions, as listed in the weaknesses below.

**Weaknesses:**

1. The experiments do not convincingly demonstrate the “actual benefit” of the proposed approach relative to the related methods. I think, specifically, the paper should have presented how $\delta$-CS enhances robustness against proxy model misspecification by empirically evaluating the performance across varying proxy model initializations (e.g., from poor to high quality). Although the hard version of the task with a small search space (TF-Bind-8) is given, the hypothesis (*) is not quantitatively validated well enough when the search space is scaled up.

2. Another major concern is the lack of proxy model ablation, i.e. how does the performance of $\delta$-CS change under different proxy models used as a reward function? (See also the question below) This could demonstrate the flexibility of the proposed search approach without depending on the proxy model type specification.

3. The paper presents $\delta$-CS as a novel off-policy search method for training GFlowNets under proxy misspecification, however, how does $\delta$-CS differ from existing off-policy search methods for GFlowNets such as [7,8]? The method [7] proposes to train GFlowNet with the exploration of local neighborhoods, similar to restricting the search space via conservativeness, and [8] incorporates Thompson sampling to improve exploration. I think these should have been direct baselines of $\delta$-CS, as it would show the effectiveness of $\delta$-conservativeness against local search and sampling starting from the same initial misspecified proxy model. ( which were first published in 2023, so I think it is a fair comparison to ask). How does $\delta$-CS differ from these methods?


4. Using BO with a generative model [5,6] is a well-established approach to biological sequence design, which would provide a more competitive baseline than classical BO. Additionally, it would have provided stronger evidence to compare against existing trust region methods for biological sequence design [1,2,3,4], as the search intuition is similar and would be appropriate baselines for the proposed method-- since $\delta$-CS constraints the search space to reliable regions, similar to the trust region concept. BO is selected as a classical baseline, however, there are state-of-the-art trust region BO methods [2,3] that apply to combinatorial search spaces such as in biological sequence design space.

5. A quantitative analysis showing how the level of conservativeness/noise injection amount differs throughout the active learning rounds, which would validate the balancing capability of $\delta$-CS.

6. A part that requires a bit of clear discussion is that, if the proxy model is not of good quality then exploration helps to recover from getting stuck at local optima and collect diverse points, whereas over-exploration yields unreliable results as explained for using GFlowNets case. I think this trade-off could have been argued more clearly while motivating the method.

7. How do convergence and diversity of sampled sequences change w.r.t the batch size? It would have been better to show batch size ablations in a problem domain.

8. The paper lacks a discussion or a theoretical analysis on how $\delta$ yields better exploration, sample complexity, and/or convergence throughout the active learning process.

9. There is no clear discussion on the limitations and potential drawbacks of the method.



**Minor:**

- Line 77: The explanation regarding the search space size is unclear. Also, why it is necessary to show that the search space size is greater than 10^309? There is no explanation of what those numbers correspond to. (It is only clear to a person who is familiar with protein design literature, that 20 represents amino acids and the exponent is the length of the GPF protein sequence.)
- Regarding writing, the transition between RL and active learning terminology could be smoother. Specifically, the term "trajectories" is used without giving an explanation. Do they refer to the trajectory of sequences generated throughout the active learning/query rounds? For instance, the sentence (lines 80-81) could have been expressed more clearly, e.g. searching sequences around the neighborhood of observed (or annotated) data points. Or sampling full trajectories (in line 88).
- Appendix B.1, line 832, misspelling in "conservatives"




> [1] Biswas, S., Khimulya, G., Alley, E.C. et al. Low-N protein engineering with data-efficient deep learning. Nature Methods 18, 389–396 (2021).

> [2] Wan, Xingchen & Nguyen, Vu & Ha, Huong & Ru, Binxin & Lu, Cong & Osborne, Michael. Think Global and Act Local: Bayesian Optimisation over High-Dimensional Categorical and Mixed Search Spaces. International Conference on Machine Learning (ICML), 2021.

> [3] David Eriksson, Michael Pearce, Jacob Gardner, Ryan D Turner, and Matthias Poloczek. Scalable Global Optimization via Local Bayesian Optimization. In Advances in Neural Information Processing Systems (NeurIPS), volume 32, pages 5496–5507, 2019.

> [4] Khan A, Cowen-Rivers AI, Grosnit A, Deik DG, Robert PA, Greiff V, Smorodina E, Rawat P, Akbar R, Dreczkowski K, Tutunov R, Bou-Ammar D, Wang J, Storkey A, Bou-Ammar H. Toward real-world automated antibody design with combinatorial Bayesian optimization. Cell Rep Methods. 2023 Jan 3;3(1):100374.

> [5] Samuel Stanton, Wesley Maddox, Nate Gruver, Phillip Maffettone, Emily Delaney, Peyton Greenside, and Andrew Gordon Wilson. Accelerating bayesian optimization for biological sequence design with denoising autoencoders.
International Conference on Machine Learning (ICML), 2022.

> [6] Nate Gruver, Samuel Stanton, Nathan Frey, Tim G. J. Rudner, Isidro Hotzel, Julien Lafrance-Vanasse, Arvind Rajpal, Kyunghyun Cho, and Andrew Gordon Wilson. 2024. Protein design with guided discrete diffusion. In Proceedings of the 37th International Conference on Neural Information Processing Systems (NeurIPS), 2023.

> [7] Kim, M., Yun, T., Bengio, E., Zhang, D., Bengio, Y., Ahn, S., & Park, J. (2023). Local search gflownets. arXiv preprint arXiv:2310.02710.

> [8] Rector-Brooks, Jarrid, et al. "Thompson sampling for improved exploration in gflownets." arXiv preprint arXiv:2306.17693 (2023).

**Questions:**

1. Can we use $\delta$-Conservative Search with any other proxy model?

2. For the baseline methods that require surrogate/proxy (e.g. BO), is the same proxy model of $\delta$-CS used, with the same initial dataset?

3. How diversity/novelty is calculated? Is it the diversity of the last top-k batches? Presenting this metric across active learning rounds would have enhanced clarity and provided a more comprehensive understanding of the diversity achieved by $\delta$-CS.

4. In Figure 5, AAV task, there is a substantial difference in diversity between the proposed method and baselines. Given that the search space is medium-sized—somewhere between RNA-A and GFP—what might be the reason for this significant disparity?

5. How does performance vary with respect to the initial dataset size, particularly in domains with large search spaces? The initial dataset sizes currently employed are very large (e.g. 50% of the whole search space for TF-Bind-8, or the value set for GFP is again too much) compared to what is used for active learning approaches in the literature.

6. How does noise injection/level of conservativeness differ throughout the active learning rounds? Have you analyzed the behavior with respect to the proxy uncertainty?

7. Line 478, Shouldn't the phrase in line 478 be "prone to generating low rewards"?

8. What is the main source of benefit of using an RL method that utilizes a proxy model (as a reward function) and trains a generative policy instead of using some other active learning frameworks that directly utilize inexpensive proxy/surrogate models that can work under very limited training data? Why train a reward model + policy network instead of directly using the proxy model for querying?

---

> ### Author Response · Authors · 2024-11-22
> **Responses (1/4)**
>
> ### W1. proxy model misspecification. the hypothesis (*) is not quantitatively validated well enough when the search space is scaled up
> ### Q1. Can we use $\delta$-Conservative Search with any other proxy model?
>
> Thank you for pointing this out. To validate our hypothesis on larger-scale tasks, we conducted additional experiments on AAV and GFP tasks (see tables below), which we have updated in the revised manuscript.
>
> **AAV with CNN**
> |  | $D_{0}$ | $D_{heldout, \leq 1}$ | $D_{heldout, \leq 2}$ | $D_{heldout}$ |
> | - | - | - | - | - |
> | Spearman $\rho$ | 0.943  | 0.872 | 0.777 | 0.407 |
>
> **GFP with CNN**
> |  | $D_{0}$ | $D_{heldout, \leq 1}$ | $D_{heldout, \leq 2}$ | $D_{heldout}$ |
> | - | - | - | - | - |
> | Spearman $\rho$ | 0.551  | 0.293 | 0.098 | -0.352 |
>
> ---
>
> ### W2. Another major concern is the lack of proxy model ablation
>
> Thanks for pointing that out. To verify our hypothesis on various types of proxy models, we conducted additional experiments using a different proxy, MuFacNet [1], in AAV, GFP, and RNA tasks (the default proxy model is based on 1d-CNN). With another proxy model, the trends are consistent, validating our hypothesis. See the results below (we have also updated this in the revised manuscript).
>
> [1] Ren, Zhizhou, et al. "Proximal exploration for model-guided protein sequence design." International Conference on Machine Learning. PMLR, 2022.
>
>
> **AAV with MuFacNet**
> |  | $D_{0}$ | $D_{heldout, \leq 1}$ | $D_{heldout, \leq 2}$ | $D_{heldout}$ |
> | - | - | - | - | - |
> | Spearman $\rho$ | 0.932  | 0.893 | 0.827 | 0.371 |
>
> **GFP with MuFacNet**
> |  | $D_{0}$ | $D_{heldout, \leq 1}$ | $D_{heldout, \leq 2}$ | $D_{heldout}$ |
> | - | - | - | - | - |
> | Spearman $\rho$ | 0.456  | 0.404 | -0.076 | -0.551 |
>
>
> **Maximum scores**
> | | RNA-A | GFP | AAV |
> | - | - | - | - |
> |CNN| 1.055 ± 0.000 | 3.592 ± 0.003 | 0.708 ± 0.010 |
> | MufacNet | 1.050 ± 0.003 | 3.592 ± 0.005 | 0.699 ± 0.017 |
>
> ---
>
> ### W3. how does δ-CS differ from existing off-policy search methods for GFlowNets such as [7,8]? (Especially, LS-GFN [7])
>
> There are many off-policy methods in GFN, as you mentioned (we have updated them in the related works section). Among them, the most relevant method is Local Search GFN (LS-GFN) [7], where both our methods focus on restricted search in local regions for reward exploitation (the other suggested method, Thompson Sampling [8] is for diverse exploration rather than conservatism).
>
> We made a direct comparison with LS-GFN:
>
> Results (MAXIMIUM)
>
> | | RNA-A (L=14)  | RNA-B (L=14)  | RNA-C (L=14)  | TFBind8 (L=8) | GFP (L=238)   | AAV (L=90)    |
> | - | - | - | - | - | - | - |
> | LS-GFN (back-and-forth search) | 0.613 ± 0.009 | 0.572 ± 0.003 | 0.722 ± 0.009 | 0.977 ± 0.008 |  3.592 ± 0.001 | 0.549 ± 0.005 |
> | Ours ($\delta$-CS)| 1.055 ± 0.000 | 1.014 ± 0.001 | 1.094 ± 0.045 | 0.981 ± 0.002 | 3.592 ± 0.003 | 0.708 ± 0.010 |
>
>
> The reason that LS-GFN performs worse than ours is that they utilize back-and-forth search using backward and forward policies, where their search space in unidirectional sequences is limited to only the leaf part of sequences. On the other hand, our local neighbors are defined by a Hamming ball of sequence space; we distribute the search not only on the leaf part but also equally on the every tokens in sequences (from head to leaf), which makes for a more flexible local search.
> More results are provided in the updated manuscript (Appendix B.5).

---

> ### Author Response · Authors · 2024-11-22
> **Responses (2/4)**
>
> ### W4.More BO baselines (especially with trust region)
>
> Thank you for your suggestion. We agree that comparison with state-of-the-art trust-region based BO methods can provide stronger evidence of the effectivenss of our method. We conduct experiment with TuRBO [1], a widely used trust-region based BO method for our setting. As shown in the table, while TuRBO exhibits generally higher score than classical BO, our method surpass TuRBO across various tasks, exhibiting the superiority of our $\delta$-CS constraints.
>
> maximum
> |       | RNA-A (L=14)  | RNA-B (L=14)  | RNA-C (L=14)  | TFBind8 (L=8) | GFP (L=238)   | AAV (L=90)    |
> | - | - | - | - | - | - | - |
> | BO    | 0.722 ± 0.025 | 0.720 ± 0.032 | 0.506 ± 0.003 | 0.977 ± 0.008 | 3.572 ± 0.000 | 0.500 ± 0.000 |
> | TuRBO | 0.935 ± 0.034 | 0.921 ± 0.052 | 0.912 ± 0.036 | 0.974 ± 0.019 | 3.586 ± 0.000 | 0.500 ± 0.000 |
> | Ours  | **1.055** ± 0.000 | **1.014** ± 0.001 | **1.094** ± 0.045 | **0.981** ± 0.002 | **3.592** ± 0.003 | **0.708** ± 0.010 |
>
> median
> |       | RNA-A (L=14)  | RNA-B (L=14)  | RNA-C (L=14)  | TFBind8 (L=8) | GFP (L=238)   | AAV (L=90)    |
> | ----- | ------------- | ------------- |:------------- |:------------- |:------------- |:------------- |
> | BO    |  0.510 ± 0.008 | 0.502 ± 0.013 |  0.506 ± 0.003 |  0.806 ± 0.007 |  3.378 ± 0.000 | 0.478 ± 0.000 |
> | TuRBO | 0.622 ± 0.046 | 0.629 ± 0.030 | 0.541 ± 0.068 | **0.974** ± 0.019 | **3.583** ± 0.003 | 0.500 ± 0.000 |
> | Ours  | **0.939** ± 0.008 | **0.929** ± 0.004 | **0.972** ± 0.043 | 0.971 ± 0.006 | 3.567 ± 0.003 |  **0.663** ± 0.007 |
>
> [1] David Eriksson, Michael Pearce, Jacob Gardner, Ryan D Turner, and Matthias Poloczek. Scalable Global Optimization via Local Bayesian Optimization. In Advances in Neural Information Processing Systems (NeurIPS), volume 32, pages 5496–5507, 2019.
>
> ---
>
> ### W5. A quantitative analysis showing how the level of conservativeness/noise injection amount differs throughout the active learning rounds
> ### Q6. How does noise injection/level of conservativeness differ throughout the active learning rounds? Have you analyzed the behavior with respect to the proxy uncertainty?
>
>
> The level of conservativeness varies depending on the data point. As active learning progresses through more rounds, it continues to seek new data points with high predictive uncertainty. Therefore, as active learning actively discovers new data points, the conservativeness level associated with these points is also dynamic. No empirical distributional tendencies regarging on the number of active rounds have been found, so the model must adapt in every active learning round and for each data point $x$.
>
> ---
>
> ### W6. A part that requires a bit of clear discussion is that, if the proxy model is not of good quality then exploration helps to recover from getting stuck at local optima and collect diverse points, whereas over-exploration yields unreliable results as explained for using GFlowNets case. I think this trade-off could have been argued more clearly while motivating the method
>
> Thanks for opening insightful discussion. Such diverse exploration can help recover from bad local optima, yet it can produce unreliable results if the exploration is too extensive. Therefore, we need to balance diversity to focus on reliable regions without being so local that we cannot escape from bad local optima. Our $\delta$ factor can be interpreted as such a balancing parameter.
>
> ---
>
> ### W7. batch size ablation
>
> Thank you for your suggestion. We conduct additional experiments with query batch size 32 and 512 (default setting is 128). The same ablation is also applied for AdaLead as a baseline. We report the results in the tables below, and we also added **Figure 15,16 in Appendix B.6** to the manuscript. The results demonstrate that our method achieves superior performance compared to AdaLead across different batch size, exhibiting robustness.
>
> maximum (bs=32)
> |         | RNA-A (L=14)  | GFP (L=238)   | AAV (L=90)    |
> | - | - | - | - |
> | AdaLead | 0.866 ± 0.049 | 3.572 ± 0.000 | 0.508 ± 0.006 |
> | GFN-AL | 0.979 ± 0.007 | 3.577 ± 0.005 | 0.533 ± 0.004 |
> | Ours    | 1.021 ± 0.033 | 3.583 ± 0.006 | 0.685 ± 0.019 |
>
> maximum (bs=512)
> |         | RNA-A (L=14)  | GFP (L=238)   | AAV (L=90)    |
> | - | - | - | - |
> | AdaLead | 1.008 ± 0.034 | 3.584 ± 0.003 | 0.656 ± 0.015 |
> | GFN-AL | 1.041 ± 0.016 | 3.593 ± 0.002 | 0.579 ± 0.007 |
> | Ours    | 1.049 ± 0.005 | 3.597 ± 0.002 | 0.707 ± 0.012 |

---

> ### Author Response · Authors · 2024-11-22
> **Responses (3/4)**
>
> ### W8. theoretical analysis
>
> Thanks for pointing out. We provided theoretical analysis on exploration range and sample complexity:
>
> 1. **Exploration range**:
>
> Let $\delta \in [0,1]$ be the exploration parameter serving as the success probability in a Bernoulli distribution for each token in a sequence of length $L$. Each token independently has a probability $\delta$ of being flipped (changed) and $1 - \delta$ of remaining the same.
>
> Define the random variable $H$ as the Hamming distance—the number of differing tokens between two sequences. The probability distribution of $H$ is:
>
> $$
> P(H = k) = \binom{L}{k} \delta^k (1 - \delta)^{L - k},
> $$
>
> where $k = 0, 1, \dots, L$. This is a binomial distribution with parameters $n = L$ and $p = \delta$.
>
> The expected value and variance of $H$ are:
>
> $$
> \mathbb{E}[H] = L\delta, \quad \operatorname{Var}(H) = L\delta(1 - \delta).
> $$
>
> A higher $\delta$ increases $\mathbb{E}[H]$, expanding the exploration region in the sequence space and promoting diversity.
>
>
> 2. **Sample Complexity:** We do not introduce additional meaningful sample complexity as a scale because the noising process involves sampling from a tractable trivial distribution (Bernoulli). The major bottleneck is sampling sequences proportional to the reward, which is intractable. However, the GFlowNet (GFN) amortizes this process using a neural network, reducing the sampling cost to the neural network's forward pass with token-by-token $O(L)$ complexity, where $L$ is the sequence length. This approach is consistent with every sequence-based decoding method in deep learning.
>
>
> 3.**Convergence Throughout**: Proving the convergence of deep learning models is notoriously difficult without making substantial assumptions, even on fixed datasets. Therefore, conducting convergence analysis on active learning that includes an inner deep-learning training process is even more challenging. Instead, this paper focused to demonstrate convergence through empirical validation across six tasks.
>
> ---
>
> ### W9. There is no clear discussion on the limitations and potential drawbacks of the method.
>
> Thank you for your feedback. We added a discussion section (Section 7), including a few notes on the limitations, in the updated manuscript.
>
> ---
>
> ### Q2. For the baseline methods that require surrogate/proxy (e.g. BO), is the same proxy model of $\delta$-CS used, with the same initial dataset?
>
> We use the same 1dCNN-based proxy (from Sinai et al., 2020) for all algorithms except GFN-AL. We implement two versions and report the better results between them. The first version is the original implementation (Jain et al. 2022) with MLP-based proxy, and the second is our own implementation with 1dCNN-based proxy. Theses are described in **Implementation details** in Section 6 and Appendix A.3.
>
> ---
>
> ### Q3.How diversity/novelty is calculated? Is it the diversity of the last top-k batches? Presenting this metric across active learning rounds would have enhanced clarity and provided a more comprehensive understanding of the diversity achieved by $\delta$-CS.
>
> Following GFN-AL (Jain et al. 2022), the diversity and novelty is calculated with top-128 at the final rounds. We added the definition of these metrics in Appendix B.8.
>
> As you suggested, we report the progress of diversity and novelty across active learning rounds in Appendix B.8.
>
> ---
>
> ### Q4. In Figure 5, AAV task, there is a substantial difference in diversity between the proposed method and baselines. Given that the search space is medium-sized—somewhere between RNA-A and GFP—what might be the reason for this significant disparity?
>
> You must mean the Figure 3. The low divrsity and novelty for the baselines is because:
>
> 1) The initial dataset is not very diverse. The diversity of top-128 samples in the initial dataset is around 5.
> 2) The queried candidates from GFN-AL or GFNSeqEditor have (relatively) low oracle scores, and thus the top-128 batch couldn't be populated by newly queried diverse and novel candidates.
>
> Compared to these baselines, our method generated both diverse and novel high-score candidates through out the active learning rounds.

---

> > ### Author Response · Authors · 2024-11-22
> > **Responses (4/4)**
> >
> > ### Q5. How does performance vary with respect to the initial dataset size, particularly in domains with large search spaces? The initial dataset sizes currently employed are very large (e.g. 50% of the whole search space for TF-Bind-8, or the value set for GFP is again too much) compared to what is used for active learning approaches in the literature.
> >
> > Thank you for your suggestion. The initial dataset size is cruical for the performance of active learning approaches, especially the search space is exponentially large. We conduct ablation studies on initial dataset size in high-dimensional tasks, AAV and GFP. We conduct experiments by varying the initial dataset size $\vert\mathcal{D}\vert=1000$. As shown in the table, our method outperforms AdaLead and GFN-Al even with a small amount of datasets.
> > Also, we've discussed about TF-Bind-8 and introduced the hard version of itm whgich has a much smaller and locally biased initial dataset in the manuscript.
> >
> >
> > **maximum scores**
> > |         | GFP (L=238)   | AAV (L=90)    |
> > | ------- |:------------- |:------------- |
> > | AdaLead | 3.568 ± 0.005 | 0.557 ± 0.023 |
> > | Ours    | 3.591 ± 0.007 | 0.704 ± 0.024 |
> >
> > **median scores**
> > |         | GFP (L=238)   | AAV (L=90)    |
> > | ------- |:------------- |:------------- |
> > | AdaLead | 3.529 ± 0.006 | 0.494 ± 0.010 |
> > | Ours    | 3.570 ± 0.008 | 0.666 ± 0.018 |
> >
> > ---
> >
> > ###  Q7. Line 478, Shouldn't the phrase in line 478 be "prone to generating low rewards"?
> >
> > You're right. We've corrected the manuscript. Thank you for pointing this out!
> >
> > ---
> >
> > ###  Q8. What is the main source of benefit of using an RL method that utilizes a proxy model (as a reward function) and trains a generative policy instead of using some other active learning frameworks that directly utilize inexpensive proxy/surrogate models that can work under very limited training data? Why train a reward model + policy network instead of directly using the proxy model for querying?
> >
> > Since these sequences exist in a high-dimensional combinatorial space, generating complex sequences by directly using the proxy model to query the oracle is intractable. This approach would require additional sampling procedures like MCMC, which are slow and suffer from issues like slow exploration and mode mixing. As a result, they may not query highly informative samples (i.e., a diverse set of queries where we have high prediction uncertainty). In contrast, RL methods like GFlowNets amortize the search process and enable fast mode mixing to efficiently generate such candidates.
> >
> > ---
> >
> > ### Minor comments
> > 1. That sentence means that "$20^{238}$, which is bigger than $10^{309}$".
> > 2. We've revised our manuscript not to use "trajectory" before defining it.
> > 3. The typo has been corrected.
> >
> > We've updated our manuscript. Thanks for the detailed feedback!

---

> > > ### Author Response · Authors · 2024-11-26
> > >
> > > Dear Reviewer WGsn,
> > >
> > > As we conclude the discussion period, we would like to ask if there is any additional information we can provide that might influence your evaluation of the paper. We believe our previous response has effectively addressed your initial concerns and clarified some points that may have been overlooked. Thank you once again for your thoughtful consideration of our work.
> > >
> > > The Authors

---

> ### Comment · Reviewer_WGsn · 2024-11-26
> **Thank you for your response; I have still concern**
>
> Thanks a lot to the authors, with the addition of ablation studies and discussions on the updated manuscript. I have the following additional comment about your response:
>
> - I am still concerned about the first key evaluation point I have stated in the weaknesses section. The paper proposes $\delta$-CS as `"novel off-policy search method for training GFlowNets designed to improve robustness against proxy misspecification"`.
> I appreciate experimenting with other proxy models, which was one of my questions, however, the key missing evaluation is in what level $\delta$-CS helps with this problem? I mean, under poor to good-quality proxy models, what's the actual contribution of $\delta$-CS addition to the GFlowNets (or to GFN-AL) regarding the empirical performance? As I have stated previously, a clear empirical demonstration for this would be to initialize proxy models of different quality (e.g using varying numbers of initial training points or specifically crafting it to be worse) and evaluating different search methods from restricted ($\delta$-CS, LS-GFN) to explorative (GFN-AL and Thompson Sampling [8]) ones. I have asked for the initial dataset size ablation. The provided ablation study (also I wonder why it is not added to the updated manuscript) compares against AdaLead, however, from those results, I cannot properly evaluate why the addition of  $\delta$-CS would help. As also pointed out the reviewer 6RU1, the main improvement might stem from the use of GFlowNets rather than from the search method itself?
>
> I believe the paper has been improved, however, still missing key points. Overall, I cannot vote for acceptance here.

---

> ### Author Response · Authors · 2024-11-27
>
> Thanks for pointing out key points that can still make our paper improved.
>
> > a clear empirical demonstration for this would be to initialize proxy models of different quality (e.g using varying numbers of initial training points or specifically crafting it to be worse) and evaluating different search methods from restricted ($\delta$-CS, LS-GFN) to explorative (GFN-AL and Thompson Sampling [8]) ones.
>
> Following your suggestion, we have conducted additional experiments to clarify the contribution of the $\delta$-CS component, and we have included these results in **Appendix B.11 of the revised manuscript**.
>
> To assess the impact of $\delta$-CS and address concerns that improvements might stem solely from using GFlowNets, we intentionally degraded the proxy model's reliability. We did this by only using the lower percentiles— 50%, 25%, and 10%— of the initial dataset based on scores. Training the proxy model on datasets with lower percentile cutoffs increases misspecification, particularly affecting higher-reward data points.
>
> Under these conditions, we compared our method with GFN-AL (which employs an explorative search strategy) and LS-GFN (which uses a restricted search approach), as you suggested. This comparison isolates the specific contribution of the $\delta$-CS component over the base GFN-AL method.
>
> The results, also presented in Appendix B.11, demonstrate that all methods experience performance degradation as the proxy model becomes less reliable, which is expected. Notably, explorative search methods like GFN-AL perform significantly worse under proxy misspecification than restricted search methods (LS-GFN and $\delta$-CS). Unfortunately, we could not include Thompson Sampling GFN (explorative method) in our experiments due to the unavailability of their source code.
>
> *Table. Maximum values with varying initial dataset quality (AAV)*
>
> | Method   | 50th       | 25th       | 10th       |
> |----------|------------|------------|------------|
> | GFN-AL   | 0.102| 0.045| 0.018|
> | GFN-AL + LS-GFN | 0.314| 0.250| 0.230 |
> | **GFN-AL + $\delta$-CS** | **0.476** | **0.463**| **0.446**|
>
> The results show that our method consistently outperforms the other baselines across all levels of initial dataset percentiles (various proxy reliability). This provides empirical evidence of the clear contribution of the $\delta$-CS component.
>
> These findings suggest that the performance improvements are not solely due to the use of GFlowNets but are significantly enhanced by our proposed search method. We believe this addresses your concern by demonstrating that the addition of $\delta$-CS indeed helps, especially when dealing with unreliable proxy models.
>
>
> > The provided ablation study (also I wonder why it is not added to the updated manuscript) compares against AdaLead, however, from those results, I cannot properly evaluate why the addition of $\delta$-CS would help. As also pointed out the reviewer 6RU1, the main improvement might stem from the use of GFlowNets rather than from the search method itself?
>
> Thank you for pointing this out. In response to your suggestion, we have added GFN baselines—specifically, GFN-AL and LS-GFN—to our ablation study on different initial dataset sizes, setting $|\mathcal{D}_0| = 1000$, which are randomly collected.
>
> We present the results below:
>
> | Method    | GFP | AAV |
> |-----------|---------|-------|
> | Adalead    | 3.568 ± 0.005 | 0.557 ± 0.023 |
> | GFN-AL    | 3.586 ± 0.006 | 0.560 ± 0.008 |
> | GFN-AL + LS-GFN    |3.580 ± 0.003 | 0.493 ± 0.006 |
> | **GFN-AL + $\delta$-CS** | **3.591 ± 0.007** | **0.704 ± 0.024** |
>
>
> As shown in the table, the inclusion of the $\delta$-CS component leads to performance improvements over both GFN-AL and LS-GFN, which are representative search methods. This indicates that the main improvement stems from the search capability of $\delta$-CS. By restricting the search to a Hamming ball in the sequence space adjusted by the parameter $\delta$, $\delta$-CS enables a more robust search even when the proxy model is inaccurate.
>
> **These results have been included in Appendix B.10 of the revised manuscript.**
>
>
> ---
>
> Thanks for your careful and detailed feedback. Please let me know if any additional concerns have not been resolved.

---

> ### Author Response · Authors · 2024-12-02
>
> Dear Reviewer WGsn,
>
> We wanted to gently remind you that our discussion period is coming to an end soon. If you feel that your concerns have been fully addressed in our previous responses, we kindly ask you to consider revising the score. Should there be any remaining issues or questions, please do not hesitate to let us know at your earliest convenience.
> Thank you for your time and consideration.
>
> Warm regards,
>
> The Authors

---

### Official Review · Reviewer_t88c · 2024-11-03

**Soundness:** 2
**Presentation:** 3
**Contribution:** 3
**Rating:** 6
**Confidence:** 3

**Summary:**

This paper introduces a novel method called $\delta$-Conservative Search ($\delta$-CS) for designing biological sequences with desired properties. Experiments across DNA, RNA, protein, and peptide design tasks demonstrate that δ-CS outperforms traditional methods, finding higher-scoring sequences. The approach successfully combines the advantages of evolutionary and reinforcement learning methods, offering a scalable and robust framework for biotechnological applications​.

**Strengths:**

This paper leverages GFlowNets for sequential generation, allowing it to efficiently generate diverse and high-reward biological sequences., and this method’s emphasis on balancing novelty with conservativeness addresses a key challenge in RL for scientific domains where exploration can easily veer into unreliable regions. This paper includes comparisons with various baselines and established benchmarks in biological sequence design.

**Weaknesses:**

1. Novelty: This paper draws from evolutionary search and existing GFlowNet frameworks, combining aspects of both rather than developing a unique algorithmic structure. Specifically, $\delta$-Conservative Search extends GFlowNets by adding a conservativeness parameter, δ, to manage exploration around known data. While useful, balancing exploration with reliability is a recurring theme in off-policy RL, particularly with techniques like uncertainty-based exploration in Bayesian optimization and upper confidence bounds (UCB).
2. Evaluating a sequence's performance is critical in the design of biological sequences. A fast, low-cost, and relatively accurate method can often significantly reduce the complexity of the problem. In this paper, the reward information provided to the model is Oracle f. Since the reward signal provided by $f$, $f$ determines the model's learning, the accuracy and generalizability of $f$ are crucial. I doubt the evaluation and experiments are testing out-of-distribution sequences, which rely on the accuracy of $f$.

**Questions:**

1. Regarding the choice of $\delta$, although the authors reported a control experiment on $\delta$ in the appendix, are all the experiments in Table 1 based on the same $\delta$?
2. What is the initial dataset $D_0$? How is it initialized? What is the cost to acquire the reward for all sequences in $D_t$ in line 3 of Alg.1?
3. It seems that model $f$ is fixed in training, and it provides reward information for $\delta$-CS; how can we ensure $f$ can provide safe and correct reward information?

---

> ### Author Response · Authors · 2024-11-22
> **Response (1/2)**
>
> ### W1: Novelty
>
> We think a proper way to combine search methods ($\delta$-CS) and learning methods (GFN) together is not trivial, as many works in biological sequence design that use exploration-exploitation balancing methods like UCB or evolutionary search-based methods like GFNSeqEditor did not get satisfactory performance compared to our method. That shows our novel approach, which controls conservativeness parameters in terms of the level of the Hamming ball of local sequence space, was more effective than others, and such integration with GFN-AL and uncertainty adaptation was critical to performance; **novel combination** of the techniques is clearly categorized as novelty in conference review guidelines.
>
> ### W2: Evaluation and experiments are testing out-of-distribution sequences, which rely on the accuracy of $f$
> ### Q3: It seems that model $f$ is fixed in training, and it provides reward information for $\delta$-CS; how can we ensure $f$ can provide safe and correct reward information?
>
> It seems there is a slight misunderstanding in this generative active learning benchmark for biological sequence design. In this setting, the **oracle** function $f$, representing the **ground truth**, can only be accessed a limited number of times. As a result, researchers focus on introducing a proxy model $f_{\theta}$ that learns to imitate the oracle function using the available dataset.
>
> Thus, the setting does not require consideration of the out-of-distribution (OOD) capability of the oracle $f$. However, we do need to account for the OOD generalization capability of the proxy $f_{\theta}$, which is highly challenging. This motivates the use of conservative search strategies when the proxy exhibits high uncertainty. Consequently, the proxy function's limited accuracy further strengthens the rationale for using the $\delta$-CS off-policy method. This method enables us to compare with existing approaches that rely more heavily on the proxy function’s performance, highlighting the novelty of our approach.

---

> ### Author Response · Authors · 2024-11-22
> **Response (2/2)**
>
> ### Q1: Regarding the choice of $\delta$, although the authors reported a control experiment on $\delta$ in the appendix, are all the experiments in Table 1 based on the same $\delta$?
>
> For the short sequence task with (e.g., DNA, RNA), we set $\delta = 0.5$, and for the long sequence task (AAV, GFP), we set $\delta = 0.05$. These parameter values were roughly tuned using some validation experiments. Specifically, we conducted tests with a proxy model trained on a portion of the initial dataset and a validation model trained on the entire initial dataset.
>
> ### Q2: What is the initial dataset $D_0$? How is it initialized? What is the cost to acquire the reward for all sequences in $D_t$ in line 3 of Alg.1?
>
> As we noted in experimental section, the initial dataset $D_0$ for TFbind, RNA, we follows exisiting benchmark [1,2]. We collect $D_0$ for AAV and GPF based on [3], by using evolutionary search on wild type sequences given from [3].
>
> The cost to acquire the reward for all sequences in $D_t$ is not required in **line 3** (the section for rank-based sampling on dataset) because they are already precomputed for each data point and included in the dataset.
>
> [1] Kim, Minsu, et al. "Bootstrapped training of score-conditioned generator for offline design of biological sequences." Neural Information Processing Systems (NeurIPS), 2024.
>
> [2] Trabucco, Brandon, et al. "Design-bench: Benchmarks for data-driven offline model-based optimization." International Conference on Machine Learning. PMLR, 2022.
>
> [3] Sinai, Sam, et al. "AdaLead: A simple and robust adaptive greedy search algorithm for sequence design." arXiv preprint arXiv:2010.02141 (2020).

---

> > ### Author Response · Authors · 2024-11-26
> >
> > Dear Reviewer t88c,
> >
> > As we conclude the discussion period, we would like to ask if there is any additional information we can provide that might influence your evaluation of the paper. We believe our previous response has effectively addressed your initial concerns and clarified some points that may have been overlooked. Thank you once again for your thoughtful consideration of our work.
> >
> > The Authors

---

> > > ### Comment · Reviewer_t88c · 2024-11-26
> > > **Thank you for the response.**
> > >
> > > I thank for the author's response and appreciate the additional experiments. I have a misunderstanding to this work. The response has addressed my concerns. Therefore, I raised the rating.

---

> > > > ### Author Response · Authors · 2024-11-26
> > > >
> > > > Thank you for updating your score to reflect acceptance!

---

### Official Review · Reviewer_AHCz · 2024-11-04

**Soundness:** 3
**Presentation:** 3
**Contribution:** 3
**Rating:** 8
**Confidence:** 3

**Summary:**

The paper studies the problem of generative search, where we seek to generate novel and diverse particles from a large design space that yield high reward. A core practical challenge is that the proxy reward model used for the generative search is unreliable, particularly away from its supporting distribution. The paper proposes a new approach, $\delta$-conservative search, that controls "how much to trust" the proxy reward model during generative search. The approach is validated with extensive evaluation and shown to outperform other methods.

**Strengths:**

* The exposition of the paper is very clear.
* $\delta$-CS which works by injecting noise and learning to denoise the signal is a neat idea to constrain generative search.
* The method is evaluated against many strong baselines on relevant tasks, and shown to outperform all baselines most of the time.

Overall, this is a good contribution to the line of work on GFlowNets.

**Weaknesses:**

* In the adaptive method for setting $\delta$, why does setting $\lambda \sigma \approx 1/L$ not set $\delta \approx \delta_{const} - 1/L$ to a constant that does not depend on the uncertainty $\sigma$? Given this, I find the name "adaptive" and the exposition slightly misleading. The adaptivity seems to stem from the sequence length $L$ and not the uncertainty about the reward proxy $\sigma$. How are the experimental results affected if $\lambda$ was chosen a constant?

* It would be nice to see the evaluation of various $\delta$ from Appendix B.1 extended to multiple datasets. It seems that the choice of $\delta$ is critical for the achieved performance and it would be interesting what ranges of $\delta$-values outperform the strongest baseline in each of the tasks to develop an understanding for the robustness of $\delta$-CS.

**Questions:**

* Can the authors elaborate how $\delta$ is chosen in the experiments? Is this a choice that can be reproduced in practice, i.e., does it require running $\delta$-CS with multiple (all?) values of $\delta$, or does it require determining $\delta$ from a (small) validation experiment, or something else?

---

> ### Author Response · Authors · 2024-11-22
> **Response**
>
> ### W1: The adaptive $\delta$ seems not to be adaptive to uncertainty, just constant.
>
> We acknowledge that our initial expression may cause confusion. Our method adapts to each datapoint $x$ (a given sequence to edit) based on the prediction uncertainty for $x$, leveraging $\sigma(x)$ to adjust $\delta$. The misleading part arose from our handwavy expression $\lambda \sigma \approx 1/L$. Therefore, we revised it in the manuscript as
>
> $$\lambda E_{P_{\mathcal{D}_0}(x)}[\sigma(x)] \approx \frac{1}{L},$$
>
> which means we set the scaling hyperparameter $\lambda$ based on the average value of $\sigma(x)$ during the first round. Consequently, $\delta \approx \delta_{\text{const.}} - 1/L$ is not true because: 1) $\sigma$ varies for each datapoint $x$'s, and 2) even for the same $x$, $\sigma(x)$ evolves throughout the active learning rounds.
>
> ### W2: It would be nice to see the evaluation of various $\delta$ from Appendix B.1 extended to multiple datasets. It seems that the choice of $\delta$ is critical for the achieved performance and it would be interesting what ranges of $\delta$-values outperform the strongest baseline in each of the tasks to develop an understanding for the robustness of $\delta$-CS.
>
> Thank you for your suggestion. We conduct additional experiments with different $\delta$ values to validate the robustness of $\delta$-CS. The summarized results are described in the tables below, and we also include more comprehensive results in Appendix B.1 of the updated manuscript.
>
> **RNA-A**
> |  | $\delta = 0.1$ | $\delta = 0.25$ | $\delta = 0.5$ | $\delta = 0.75$ | GFN-AL |
> | - | - | - | - | - | - |
> | Max | 1.040 ± 0.014 | 1.031 ± 0.026 | **1.055** ± 0.000 | 0.925 ± 0.055 | 1.030 ± 0.024 |
> | Median | 0.918 ± 0.034 | 0.916 ± 0.028 | **0.939** ± 0.008 | 0.776 ± 0.018 | 0.838± 0.013 |
>
>
> **GFP**
> | | $\delta = 0.01$ | $\delta = 0.025$ | $\delta = 0.05$ | $\delta = 0.075$ | GFN-AL |
> | - | - | - | - | - | - |
> | Max | 3.593 ± 0.003 | **3.594** ± 0.005 | 3.592 ± 0.003 | 3.592 ± 0.005 | 3.578 ± 0.003 |
> | Median | 	**3.578** ± 0.002 | 3.574 ± 0.003 | 3.567 ± 0.003 | 3.563 ± 0.004 | 3.511 ± 0.006 |
>
> ---
>
> ### Q1: Can the authors elaborate how $\delta$ is chosen in the experiments? Is this a choice that can be reproduced in practice, i.e., does it require running $\delta$-CS with multiple (all?) values of $\delta$, or does it require determining $\delta$ from a (small) validation experiment, or something else?
>
> We set $\delta = 0.5$ for short sequence tasks (e.g, DNA, RNA), and $\delta = 0.05$ for long sequence tasks (e.g., GFP, AAV). This hyperparameter was very roughly tuned with small validation experiments; we conducted tests with a proxy model trained on just part of the initial dataset and a validation model trained on the entire initial dataset as oracle. These experiments were only done in the short DNA sequence task and the long protein sequence task. This tuning can be further improved if we finely tune task-by-task using such validation experiments.

---

> > ### Comment · Reviewer_AHCz · 2024-11-25
> >
> > Thank you for the clarification of the adaptive approach. I still feel that the reliance on the hyperparameter $\delta$ is one of the main limitations of the proposed method, as also pointed out by reviewer 6RU1. The ablations in B.1 show that the method is somewhat sensitive to the choice of $\delta$, and it might be useful to emphasize practical means of choosing $\delta$ in the manuscript.
> >
> > Overall, I nevertheless feel like this paper is a good contribution and am inclined to keep my positive score.

---

> > > ### Author Response · Authors · 2024-11-26
> > >
> > > Thank you for continuing to support our paper with a positive high score. As you suggested, we will include a practical choice of $\delta$ in the manuscript based on our discussions.

---

### Author Response · Authors · 2024-11-22
**General response**

We are sincerely grateful to the reviewers for their valuable feedback on our manuscript. We have addressed each comment individually and revised the manuscript accordingly; due to space constraints, we have moved the pseudo-code to Appendix A. Additionally, we have conducted further experiments to verify our method more robustly. Please review the revised manuscript along with our responses.

---

### Meta-Review · Area_Chair_i8VE · 2024-12-20

**Metareview:**

This paper propose a new robust RL method to combat reward misspecification focusing on the biological sequence design task. There are a lot of back and forth discussion between the author and reviewers.

The main drawback of the paper lies in its inadequate experimental evaluation and a lack of competitive baselines. These are partially improved during the rebuttal phase with new experiments. However, what remains to be a problem, as raised by both reviewer WGsn and reviewer 6RU1, is that because GFN+$\delta$-CS is the only baseline that incorporate the proposed $\delta$-CS method, it is unclear whether the benefit comes from $\delta$-CS or from the unique combination of GFN+$\delta$-CS. It could be that $\delta$-CS only work well with GFN but not with any other base algorithm. Given that this is a pure empirical paper, the current experimental evaluation is inadequate to support its claim.

I suggest that the author follow the suggestion of reviewer WGsn and 6RU1 in designing informative experiments.

**Additional Comments On Reviewer Discussion:**

NA

---

### Decision · Program_Chairs · 2025-01-22

Reject